

# Sources of Uncertainty in the Global Fire Model SPITFIRE: Development of LPJmL-SPITFIRE1.9 and Directions for Future Improvements

Luke Oberhagemann[1,2], Maik Billing[2], Werner von Bloh[2], Markus Drüke[2,3], Matthew Forrest[4], Simon P.K. Bowring[5], Jessica Hetzer[4], Jaime Ribalaygua Batalla[6], and Kirsten Thonicke[2]

[1]University of Potsdam, Karl-Liebknecht Str. 24/25, Potsdam, Germany
[2]Potsdam Institute for Climate Impact Research, Telegrafenberg, Potsdam, Germany
[3]Deutscher Wetterdienst, Hydrometeorologie, Frankfurter Str. 135, 63067 Offenbach, Germany
[4]Senckenberg Biodiversity and Climate Research Center, Senckenberganlage 25, 60325, Frankfurt, Germany
[5]Laboratoire des Sciences du Climat et de l'Environnement (LSCE), IPSL-CEA-CNRS-UVSQ, Université Paris-Saclay, Gif-sur-Yvette, France
[6]Meteogrid, Almansa 88, Madrid, Spain

**Correspondence:** Luke Oberhagemann (oberhagemann@uni-potsdam.de)

**Abstract.** Since its development in 2010, the SPITFIRE global fire model has had a substantial impact on the field of fire modelling using dynamic global vegetation models. It includes process-based representations of fire dynamics, including ignitions, fire spread and fire effects, resulting in a holistic representation of fire on a global scale. Previously, work has been undertaken to understand the strengths and weaknesses of SPITFIRE and similar models by comparing their outputs against
remotely sensed data. We seek to augment this work with new validation methods and extend it by completing a thorough review of the theory underlying the SPITFIRE model to better identify and understand sources of modelling uncertainty. We find several points of improvement in the model, the most impactful being an incorrect implementation of the Rothermel fire spread model that results in strong upward biases in fire rate of spread, and a live grass moisture parametrization that results in substantially too low live grass moisture contents. The combination of these issues leads to excessively large and intense
fires, particularly on grasslands, that bias SPITFIRE toward high tree mortality. We resolve these issues by correcting the implementation of the Rothermel model and implementing a new live grass moisture parametrization, in addition to several other improvements, including a multi-day fire spread algorithm, and evaluate these changes in the European domain. Our model developments allow SPITFIRE to incorporate more realistic live grass moisture contents, and result in more accurate burnt area on grasslands and reduced tree mortality. This work provides a crucial improvement on the theoretical basis of the SPITFIRE
model, and a foundation upon which future model improvements may be built. In addition, this work further supports these model developments by highlighting areas in the model where high amounts of uncertainty remain, based on new analysis and existing knowledge about the SPITIFRE model, and identifying potential means of mitigating them to a greater extent.



## 1 Introduction

Fire is an important component of the earth system, influencing global carbon cycles and modifying vegetation (e.g. Archibald
et al., 2018; Bowman et al., 2009). Because of these global scale impacts, Dynamic Global Vegetation Model (DGVM)-based
fire models have been developed with the goal of simulating fire on a global scale and incorporating vegetation feedbacks in
predictions of future fire regimes. One such model, the SPread and InTensity of FIRE (SPITFIRE) global fire model was first
introduced by Thonicke et al. (2010). It is designed for use with DGVMs, and models fire processes on coarse temporal and
spatial scales, most often at a 0.5° by 0.5° grid resolution, and daily time steps. It has been implemented in several DGVMs
since its development for the Lund-Potsdam-Jena (LPJ) model, including the LPJ managed Land model (LPJmL), LPJ-GUESS,
ORCHIDEE and JS-BACH (Schaphoff et al., 2018a; von Bloh et al., 2018a; Lasslop et al., 2014; Lehsten et al., 2009, 2015;
Yue et al., 2014, 2015). In addition to these direct implementations of the SPITFIRE model, the derived model LPJ-LMfire
was developed in 2013 by Pfeiffer et al. (2013), and the derived model LPX was developed by Prentice et al. (2011). Ward
et al. (2018) adapted the crown scorch component of SPITFIRE for a crown fire parametrization in the fire mdel LM3-FINAL.
Detailed information on each model can be found in their respective basis papers, and a summary is available in Rabin et al.
(2017).

The SPITFIRE model has been used for a number of model application studies including, e.g.: Wu et al. (2015), comparing
LPJmL-SPITFIRE and LPJ-GUESS-SIMFIRE in the European domain, Drüke et al. (2023), examining the impact of fire on
Amazon forest regrowth under future climate change using LPJmL-SPITFIRE, Felsberg et al. (2018), examining the impact of
lightning data temporal resolution on JSBACH-SPITFIRE model results, and Hantson et al. (2015), using JSBACH-SPITFIRE
to model anthropogenic influences on global fire size distributions. LPJ-LMfire has also been widely used for studies including,
e.g.: Boulanger et al. (2022), analysing future performance of tree species in Québec, Canada, Chaste et al. (2018), comparing
model results to observations in eastern boreal Canada, Emmett et al. (2021) developing a local-scale model based on LPJ-
LMfire, and Kaplan et al. (2016), applying LPJ-LMfire to a historical study of Europe during the Last Glacial Maximum.
Due to its widespread application in global fire modelling, SPITFIRE and its derived models form the basis of a substantial
proportion of the models used for the Fire Model Intercomparison Project (FireMIP), with 4 of the 11 models used for the first
phase of the project being SPITFIRE-based (Rabin et al., 2017). The FireMIP ensemble of models, in various configurations,
has also been used in a number of studies including, e.g.: Hantson et al. (2020), analyzing the performance of FireMIP models,
Li et al. (2019), using the model ensemble to simulate historical fire emissions, Lasslop et al. (2020), studying the effect of fire
on tree cover in the FireMIP models, Andela et al. (2017), showing that FireMIP models generally do not represent the decline
in burnt area observed in satellite data, Forkel et al. (2019a), comparing burnt area drivers in FireMIP models to a random
forest model, and Teckentrup et al. (2019), performing sensitivity analyses on the FireMIP models.

The SPITFIRE model has therefore had, and continues to have, a substantial impact on the field of predictive fire modelling
on a global scale. Understanding the sources of uncertainty in the model is therefore important for contextualizing work done
with the model itself or with ensembles in which the model plays a substantial role. Many of the studies discussed analyze these
sources of uncertainty using an a posteriori approach, in which the results of model simulations are interpreted in the context of





satellite observations (see in particular Forkel et al., 2019a; Hantson et al., 2020; Andela et al., 2017; Teckentrup et al., 2019). Here, we supplement and extend this work by examining the fundamental components of the SPITFIRE model in the context of the theory on which they are based. By returning to the underlying theory, and examining its importance at DGVM scales, we help fill a critical research gap between those examining fire at local scales, and those undertaking fire modeling at a global scale.

The sources of uncertainty in the SPITFIRE model can broadly be grouped into four classes: inaccuracies in the implementation of previous research into the model, unrealistic results from equations designed for the model, general modelling simplifications, and uncertainty associated with model inputs. In the first two cases, we analyze the nature and impacts of these sources of uncertainty and provide potential solutions or road maps toward solutions. In the third and fourth cases we provide a general discussion of the nature of these simplifications to provide a better understanding of use cases for SPITFIRE. We focus in particular on two major issues that we identify in the model: an incorrect implementation of the Rothermel fire spread model and unrealistically low live grass moisture contents. This paper is organized into a brief description of the model, the results of several validation tests, an identification of sources of uncertainty in each component of the model, an improved model version in the European domain, and an outlook discussing the current status of SPITFIRE with a road map for future developments. The results shown here also focus on SPITFIRE embedded in the LPJmL DGVM, in particular versions 4 and 5.7, and on the European domain. Because of this, some of the more detailed technical changes apply more specifically to the LPJmL implementation, while broader discussions of the equations underlying SPITFIRE are more general. We have highlighted where we are aware of differences in SPITFIRE implementations within other DGVMs that alter the impacts of certain parametrizations.

## 2 Methods

### 2.1 Model structure

SPITFIRE is a holistic fire model that models fire occurrence, spread, and effects. The structure of the model is shown in Figure 1. We provide a brief introduction to the model here, for a detailed description see the referenced work.

The version of SPITFIRE forming the basis of this work is SPITFIRE embedded in the LPJmL version 4.0 DGVM, i.e. the most recently published global version of SPITFIRE (Schaphoff et al., 2018a). We use this model version as the starting point of our work because it represents the published status of LPJmL-SPITFIRE before the changes made here. This DGVM divides global vegetation into 11 Plant Functional Types (PFTs). These are divided into 3 tropical, 4 temperate and 4 boreal types. The 3 tropical PFTs are: TrBE – Tropical Broadleaved Evergreen trees, TrBR – Tropical Broadleaved Raingreen trees, and TrH – Tropical Herbaceous, i.e. tropical grasses. The 4 temperate PFTs are: TNE – Temperate Needleleaved Evergreen trees, TBE – Temperate Broadleaved Evergreen trees, TBS – Temperate Broadleaved Summergreen trees, and TH – Temperate Herbaceous, i.e. temperate grasses. The boreal PFTs are: BBS – Boreal Broadleaved Summergreen trees, BNS – Boreal Needleleaved Summergreen trees, BNE – Boreal Needleleaved Evergreen trees, and PH – Polar Herbaceous, i.e. polar grasses. In a given grid cell, each PFT is assigned a uniform size (in all dimensions including height and diameter), corresponding to an average



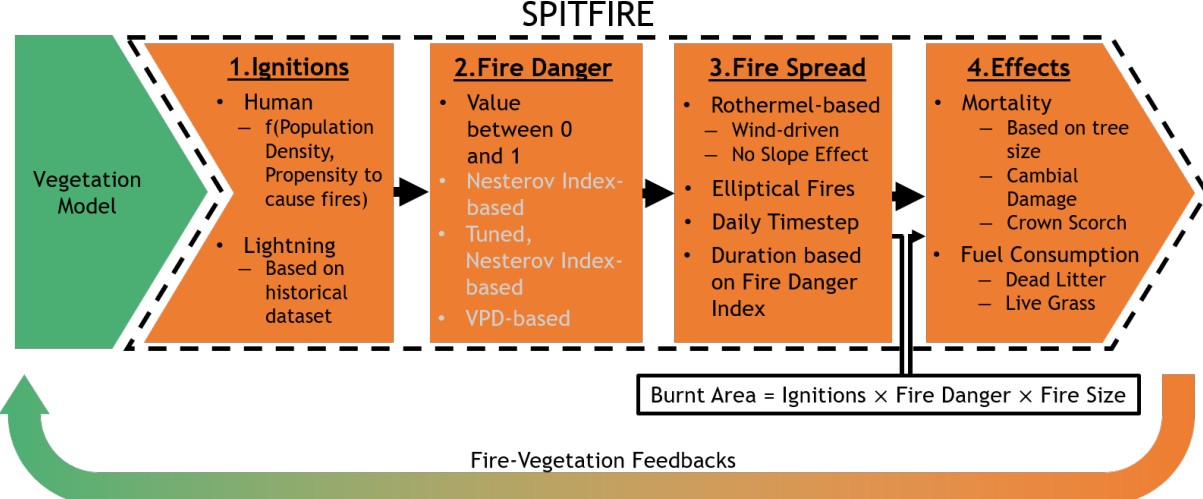

**Figure 1.** Structure of the SPITFIRE model. Grey writing indicates alternate versions of the fire danger index.

individual of that PFT, and all PFTs contribute to the same, grid cell averaged fuel bed. Note that this description does not
apply to other vegetation models in which SPITFIRE is implemented.

For new model developments, we use the most recent version of LPJmL – version 5.7, to incorporate the most recent updates (Wirth et al., 2024). The major differences between this version and version 4 are the implementation of the global nitrogen cycle, originally developed for LPJmL version 5.0, a description of which can be found in von Bloh et al. (2018a), and the implementation of a new litter parametrization that includes a new litter layer, which decomposes over time and is
renewed through vegetation mortality and leaf turnover (Lutz et al., 2019). This component is discussed in further detail in Sect. 3.2.5. The model code for versions 4 and 5 of LPJmL are available from Schaphoff et al. (2018b) and von Bloh et al. (2018b) respectively. Our simulations include human land use, which mainly affects fire in LPJmL through determining the total amount of natural land within a grid cell.

As shown in Figure 1, modelled ignitions are scaled by the fire danger, spread according to the fire spread component,
and impact vegetation according to the fuel consumption and mortality components. Component 1 of the SPITFIRE model determines the number of ignitions that may potentially become spreading fires in a grid cell on a given day. These ignitions originate from two sources: 1) human-caused ignitions that are calculated using a function of population density and the tendency of the population of a given grid cell to start fires, and 2) lightning-caused ignitions that are based on a stationary dataset. The lightning ignition component of the model assumes that 20 % of lightning strikes are cloud-to-ground and 4 % of
these cloud-to-ground strikes cause ignitions.

Subsequently, in component 2, the number of ignitions is scaled using a Fire Danger Index (FDI) with a value between 0 and 1. There are three versions of this FDI that have been implemented in different versions of SPITFIRE. The original version, described in Thonicke et al. (2010), is based on the ratio of the moisture content of the dead fuel to the moisture of extinction. It



is assumed that the probability that an ignition becomes a spreading fire is a linear function of this ratio. The dead fuel moisture
is calculated as an exponential function of the Nesterov Index and a set of mass-weighted scaling parameters that depend on
the surface-area-to-volume ratio of the 1, 10, and 100 hour fuel classes.

This original parametrization was modified in Schaphoff et al. (2018a), with the surface-area-to-volume ratio dependent
parameters replaced by tuning parameters that have separate values for each PFT. Finally, Drüke et al. (2019) introduced an
FDI parametrization based on the vapour pressure deficit (VPD) that combines the parametrization of Pechony and Shindell
(2009) with PFT-specific tuning parameters. Both updates retain the original, fuel class-weighted Nesterov parametrization for
the dead fuel moisture content from Thonicke et al. (2010), only applying the changes to the FDI.

The fire spread component, component 3, of SPITFIRE is based on the Rothermel fire spread model (Rothermel, 1972)
with modifications by Albini (1976). The fuel loads required for the Rothermel model are provided by the DGVM in which
SPITFIRE is embedded, fuel bulk densities are a function of PFT-specific values, and the surface-area-to-volume ratio of the
fuels within a fuel class is assumed to be uniform across all PFT's. Live fuel moistures are calculated as a function of modelled
soil moisture. The impact of slope on fire spread is neglected and the average daily wind speed across the grid cell is used to
calculate the rate of spread. Each fire lasts for a duration that is calculated as a function of the FDI. All fires are assumed to
be elliptical, with a size given by the rate of spread, the fire duration, and a wind-speed dependent length to breadth ratio on a
particular day in a grid cell. The burnt area for a given day is thus calculated as a product of the number of ignitions, the FDI,
and the fire size for that grid cell and day (all fires on a given day in the same grid cell have the same input parameters and,
therefore, size).

The effects of this burnt area on simulated vegetation, shown in component 4, is twofold: a portion of the modelled litter
bed and live fuel is consumed due to the fire, and modelled trees may undergo mortality due to either cambial damage or
crown scorch. modelled bark thickness, heights, and crown base heights are used in these calculations, generally calculated
dynamically by the coupled vegetation model. The probability of mortality due to cambial damage is calculated based on the
bark thickness and on modelled fire characteristics. The probability of mortality due to crown scorch is calculated based on the
height at which calculated flames have an impact on tree crowns, the modelled crown base and top heights, an assumed cylin-
drical crown structure, and PFT-specific resistance parameters. The two probabilities of mortality are combined by assuming
that they are independent. The altered fuel beds and vegetation, in turn, impact subsequent simulated fires.

## 2.2 Model forcing data

The solar radiation, precipitation, temperature, wind, and humidity data required to run the model at the 0.5° scale were
obtained from the WFDE5 bias-adjusted ERA5 data set (Cucchi et al., 2020). For the 0.07° European domain runs we use
ERA5 land data regridded to the FirEUrisk grid (Muñoz-Sabater et al., 2021; Chuvieco et al., 2023). Lightning data is derived
from the the LIS/OTD monthly, historical lightning data set (Christian, 2003), interpolated to a daily time step as described in
Thonicke et al. (2010). Population density for these runs was obtained from the HYDE 3.1 data set (Klein Goldewijk et al.,
2010).





### 2.3 New validation methods

Validation of DGVM-based fire models often involves comparisons of maps and time series of model-computed burnt area to burnt area products from satellite-based datasets. We conduct this comparison for LPJmL-SPITFIRE using the GFED4s dataset
(Randerson et al., 2015). In addition, we develop two new methods for validating SPITFIRE results. In the first, we split the global burnt area into grid cells that are dominated by tree PFTs and those that are dominated by grass PFTs, i.e. their Foliar Projective Cover (FPC) covers over half of the cell. In the case of the modelled burnt area we use the modelled FPC for this division, and in the case of the satellite-based data we divide the burnt area using the PFT distributions determined by Forkel et al. (2019b). This allows us to gain additional insight into the modelled results by examining whether the global burnt area in
the model arises from similar fire-vegetation dynamics as the observed data.

The total burnt area calculated by the model is the result of several model components, as shown in Figure 1, and this poses a challenge for determining where errors in the burnt area arise. For example, halving the number of ignitions and doubling the fire size results in the same final burnt area. To gain additional insight into the fire spread component, we introduce a second new validation method. For this, we add a feature to LPJmL-SPITFIRE that allows it to operate using prescribed fire
starts (we use the term fire starts here to distinguish these successful ignitions that result in steady-state fire spread from the total modelled ignitions, i.e. these fire starts are not reduced by the fire danger index, whereas the ignitions are). We use the Global Fire Atlas dataset and start fires in the same grid cell, and on the same day as the fires in the Global Fire Atlas (Andela et al., 2019). We then compare the burnt area calculated by aggregating all of the fires in a given grid cell from the Global Fire Atlas to the burnt area calculated by LPJmL4-SPITFIRE using these prescribed fire starts. By doing so we circumvent the
considerable uncertainty due to the modelling of ignitions and fire danger to focus on the factors that affect the rate of spread and, therefore, fire size.

## 3 Results and discussion

### 3.1 Examination of LPJmL4-SPITFIRE using new validation methods

LPJmL4-SPITFIRE shows a reasonable agreement in terms of annual global burnt area with the satellite-derived data set GFED
4s (Randerson et al., 2015), as shown in panel c) of Figure 2. Spatially, shown in panels a) and b), there is some disagreement with the validation data, with, e.g., India experiencing substantially too high burnt area, while burnt area in Australia and at high northern latitudes is underestimated (perceptually uniform colourmaps in this paper were created using colorcet, see Kovesi, 2015). Other spatial patterns, including the large burnt areas in parts of Africa are reasonably well captured.

The validation using burnt area split by PFT is shown in panels d) and e) in Figure 2. Note that the sum of the two panels d)
and e) is not the total burnt area in panel c) because many grid cells where there is non-negligible burnt area contain a substantial proportion of managed land in addition to the natural fraction, resulting in neither trees or grasses covering over half of the grid cell. These time series reveal a substantial bias in the manner in which the global burnt area is distributed. Overall, the amount of burnt area in grid cells dominated by trees agrees approximately with validation data. However, grid cells dominated



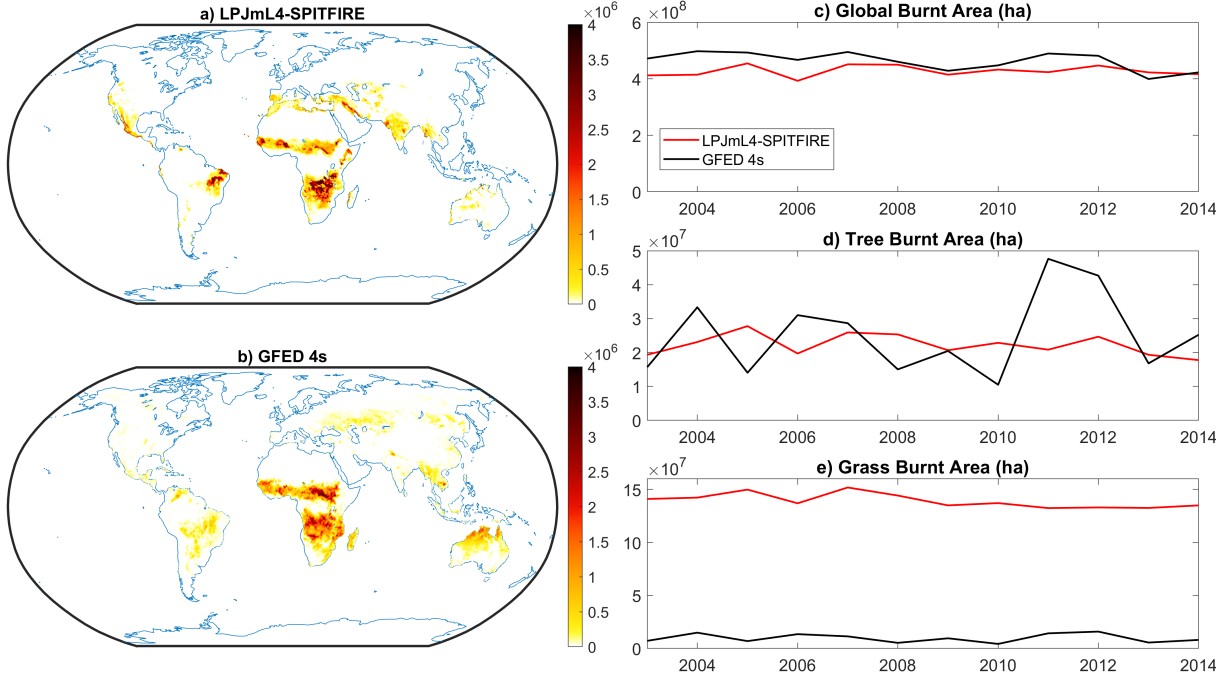

**Figure 2.** Comparison of LPJmL4-SPITFIRE burnt area with GFED 4s validation data. Maps of burnt area in panels a)-b) show a general alignment in assigning large portions of the global burnt area, in ha, to the African continent. However, there are regions of substantial geographic disagreement, particularly in India and Australia. The time series plot in panel c) shows a reasonable agreement in global annual burnt area between modelled and validation data sets. Comparisons of burnt area in grid cells that area over 50 % covered by tree PFTs in panel d show reasonable agreement, although this is strongly reduced when examining tree PFTs individually (not shown). A comparison of burnt area in grid cells that are over 50 % covered by grass PFTs in panel e) shows a strong excess in modelled burnt area.

by grasses show an extreme upward bias in burnt area. Therefore, the burnt area, while showing some agreement on a global

scale, does not appear to arise from the correct physical mechanisms.

The validation using prescribed firestarts is shown in Figure 3. The model, panel b), calculates much lower burnt area given the prescribed fire starts than the cumulative burnt area of the respective fire sizes, shown in panel a). The only minor exception is a region in southern Africa, visible also as a small region of greater burnt area in the model in panel c). Andela et al. (2019) identify this region as having a high ignition density and small fire sizes, showing that the current parametrization of

LPJmL4-SPITFIRE only reproduces burnt areas that occur due to a large number of small fires (see Figure 8 in Andela et al., 2019). Motivated by these differences, we thoroughly examine the model structure to identify sources of uncertainty that may contribute to these issues.





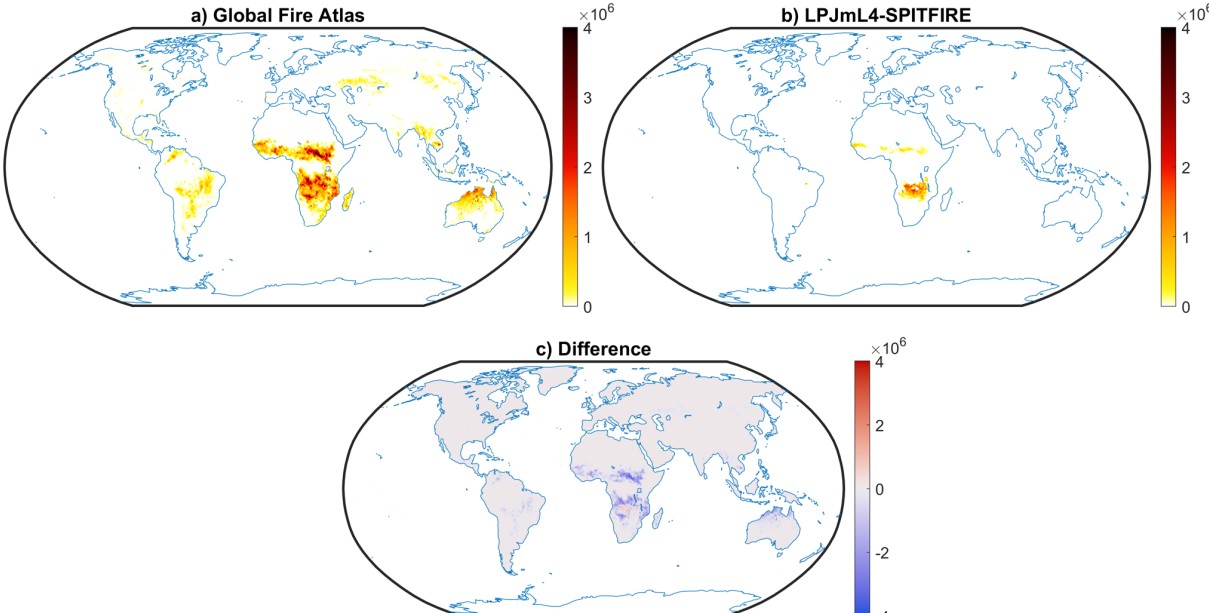

**Figure 3.** Comparison of burnt area per grid cell in the LPJmL4-SPITFIRE model, using prescribed fire starts from the Global Fire Atlas, to the burnt area per grid cell from the observed fires corresponding to those fire starts (panels b) and a) respectively). Given the same fire starts LPJmL4-SPITFIRE produces a substantially lower burnt area across the vast majority of regions. The difference map in panel c, showing the modelled burnt area minus the validation data, further illustrates this.

## 3.2 Improvements on errors and uncertainties in SPITFIRE

Two substantial errors in SPITFIRE are an incorrect weighting of parameters in the Rothermel (1972)-based rate of spread calculation that results in unrealistically large and severe fires, and unrealistically low modelled live grass moistures. In combination with the incorrect weighting factors, this can cause a strong bias towards simulated fires on grasslands as opposed to forests. In some cases, e.g. due to model tuning, other parts of the model can compensate for these errors somewhat, but this balancing of errors is often insufficient to overcome the biases introduced.

### 3.2.1 Errors in the implementation of the Rothermel Model

As described in Sect. 2.1, the rate of spread calculation, which implements the widely used Rothermel model, is a core component of SPITFIRE. We discovered several logical discrepancies between the Rothermel model, given in the original publication by Rothermel (1972) and updated by Albini (1976), and its implementation in SPITFIRE. These are described in detail in Appendix A. The main discrepancies arise from the manner in which different fuel components, separated by size and by dead or living status, are treated when they are combined in the model. Rothermel (1972) and Albini (1976) introduced weighting schemes based on the contribution of each component to the overall surface area of the fuel bed, and treat dead and live fuels





separately. Contrary to this, SPITFIRE uses a weighting scheme based on contributions to the overall mass of the fuel bed or, in the case of fuel loads, neglects the weighting factors entirely, allowing for values up to three times the desired amount. The fuel classes are then combined into a version of the Rothermel model designed for uniform fuels, rather than the non-uniform fuels present, and that, therefore, does not treat living and dead fuels separately, despite their substantially different characteristics.

To analyze the impact that these errors in the application of the Rothermel equation have on SPITFIRE model results, we compare the output of the Rothermel equation with and without errors. For this purpose, we implement the rate of spread calculations from SPITFIRE independently from LPJmL ("offline") using MATLAB (The MathWorks Inc., 2021), and develop an additional code that correctly implements the Rothermel equation for non-uniform fuel beds, which we verify by reproducing several plots in Scott and Burgan (2005). We take the intermediate step of implementing this in MATLAB as it allows us to

examine the behaviour of the SPITFIRE implementation of the Rothermel model in a much more efficient way than conducting large amounts of model runs with specified inputs, an application for which the SPITFIRE code was not set up. This MATLAB implementation also allows us to further verify our understanding of the Rothermel model, and to produce results for specific fuel and wind speed configurations that we can use for model verification of the new SPITFIRE version, described in further detail later in this section.

To compare the two approaches, we focus on the rate of spread, as this is the main output of the Rothermel model; the fire size, as the sum of fire sizes makes up the burnt area, an important output of SPITFIRE; and the scorch height, the distance from the ground that flames have an effect on the vegetation, a major factor in determining vegetation mortality. In SPITFIRE, the probability of mortality due to cambial damage is set to the fraction of a tree crown for a given PFT that is below this scorch height. For further details see Sect. 3.2.6. To compare the fire size between the Rothermel and SPITFIRE approaches,

we use the elliptical fire spread approach from SPITFIRE, and to compare the scorch height we calculate the fireline intensity using the approach in Albini (1976). This approach is slightly different from the fuel consumption-based approach used in SPITFIRE. However, we adopt it here for the sake of a more direct comparison. The equation for scorch height, in m, is given by:

$$\text{SH} = F \times I_B^{2/3}, \tag{1}$$

where, $I_B$ is the fireline intensity, in $\text{kW m}^{-1}$, and $F$ is a PFT-specific scaling parameter.

We conduct this comparison for three Scott and Burgan fuel models that capture different balances between live and dead fuels (Scott and Burgan, 2005). The Scott and Burgan fuel models are a set of parameters that describe various types of fuel beds in a manner that can be input into the Rothermel equation. The three we use in this work are TL3, moderate load conifer litter, which includes only dead fuels; TU2, moderate load humid climate timber-shrub, which includes dead and living fuels;

and GR6, moderate load humid climate grass, which is dominated by live fuels. The choice of these fuel models was also motivated by the work of Aragoneses et al. (2022) who found that these are the most widespread fuel models across Europe of the timber litter, timber understory, and grass fuel model types. Following the plots in Scott and Burgan (2005), we set the live fuel moistures to 0.6 for the herbaceous fuels and 0.9 for the woody fuels. For dead fuel moisture contents we reproduce the





scenarios in Scott and Burgan (2005), where the very low fuel moisture scenario has moistures of 3 %, 4 %, and 5 % for the 1-,
10-, and 100 hour dead fuel classes, and each subsequent scenario increases these moisture contents by 3 percentage points.

The comparison between the SPITFIRE and Albini (1976) implementations of the Rothermel equation for fuel models TL3
and TU2 are shown in Figure 4. Subplots a) and b) show the impact of the errors on the difference between the SPITFIRE rate
of spread and the correct rate of spread, with the SPITFIRE rate of spread being generally higher at low wind speeds and lower
at higher wind speeds. The effect of this on fire size is shown in subplots c) and d). Because the rate of spread difference at low

wind speeds is very large compared to the rate of spread, there is a very high peak, up to 2000 % for TL3 and 900 % for TU2,
at lower wind speeds in terms of the percent difference ($\frac{\text{SPITFIRE}-\text{correct}}{\text{correct}} \times 100\,\%$) between the SPITFIRE implementation
and the correct one. This difference becomes slightly negative at high wind speeds. While the most extreme difference is only
observed at low wind speeds, this may still pose a substantial problem for the model, as the daily and grid cell averaged wind
speeds that are used as inputs are often in this lower range (the impact on actual model results can also be seen in Sect. 3.3).

This impact of the errors in the SPITFIRE Rothermel implementation on the rate of spread and sub-components of the
Rothermel model also cause a substantial bias in the scorch height. As shown in panel e) of Figure 4, in the TL3 model this
reaches values up to 1900 % and, in contrast to the impact on fire size, remains high for all values of wind speed shown, with a
minimum percent difference for the wind speeds we analyze of about 400 %. This same pattern, with somewhat lower biases, is
also visible for the TU2 model in panel f). To illustrate this more tangibly we have calculated the difference between the scorch

heights by setting the $F$ factor in Equation 1 to 0.1. This value is within 0.01 of the values for all PFTs aside from Tropical
Broadleaved Evergreen trees, Tropical Broadleaved Raingreen trees, and Temperate Broadleaved Evergreen trees, which have
values of 0.149, 0.061, and 0.371 respectively. As shown in panels g) and h), the difference in scorch height rises rapidly,
reaching several meters at at wind speeds below $2\ \mathrm{km\,h^{-1}}$ for both of the fuel models shown. This strong bias towards high
scorch heights is likely to lead to excessive tree mortality, as further shown in comparative model runs in Sect. 3.3.

In addition to these upward biases, there is an additional dynamic that can be seen by comparing the fuel models in Figure
4. Namely, the upward bias on fire size decreases with increasing presence of live fuel. TL3, which contains no live fuels
has the highest biases and TU2, which contains some live fuels has lower biases. This is to the extent that the mostly live
GR6 fuel model, shown in the supplement in Figure S1, permits no fire spread at all. Therefore, realistic live grass moistures
should often lead to complete fire extinction on grasslands in SPITFIRE. The reason why there is still fire on grasslands in

LPJmL4-SPITFIRE despite this extreme damping is due to extremely low modelled live grass moistures, which we discuss in
the subsequent section.

To rectify these issues, we have rewritten the implementation of the Rothermel equation in the SPITFIRE model to ensure
that the SPITFIRE rate of spread calculations match those in Albini (1976). Due to the large amount of space that a detailed
description of the Rothermel model would require, we have included only the equations from the Rothermel model relevant to

the discussion above in Appendix A. For a full overview of the model we recommend Andrews (2018).

We have verified the corrected implementation by reproducing selected results in Scott and Burgan (2005). Specifically,
we tested the model with several runs for each of the TL3, TU2, and GR6 models, conducted using both the MATLAB
implementation of the Rothermel model that was, in turn, tested against Scott and Burgan (2005) and the new, corrected







**Figure 4.** Comparison between outputs of the Rothermel equation using the approach in SPITFIRE and the standard approach. Differently coloured lines indicate different dead fuel moisture contents, following the moisture content scenarios of Scott and Burgan (2005). For all plots the herbaceous live fuel moisture content is set to 60 % and the woody live fuel moisture is set to 90 %.

implementation of the Rothermel model in the SPITIFRE code. In all cases, we use the low dead fuel moisture scenario, with a

live herbaceous fuel moisture of 60 %, a live woody fuel moisture of 90 %, and the wind speed limit from Andrews et al. (2013). The full results of this comparison can be found in Table S1 in the supplement. The highest difference in terms of rate of spread that we find is 2 % and the highest difference in terms of Fireline intensity is 3 %. These differences occur in the TL3 fuel model at a wind speed of $300 \, \mathrm{m \, min^{-1}}$, which produces a relatively low intensity and rate of spread compared to the other fuel models tested. These differences amount to only $0.04 \, \mathrm{m \, min^{-1}}$ in the rate of spread and $2.67 \, \mathrm{kW \, m^{-1}}$ in the fireline intensity.





These small differences can most likely be attributed to rounding errors in the model code. The updated implementation of the Rothermel model in SPITIFRE therefore performs accurately. For further technical details on the implementation of this parametrization in the model, please consult the model code provided with this article, and we also provide the MATLAB implementation of the Rothermel model.

### 3.2.2 Live fuel moisture parametrization

LPJmL-SPITFIRE only calculates fire spread in live herbaceous fuels, this section therefore focuses on the live grass moisture parametrization contained in the model.

**Sources of error**

As alluded to in the previous section, and shown in Figure S1, the manner in which the live and dead fuel moistures are combined in SPITFIRE leads to a combined $M_f$ that is often quite high. This is to the extent that it can be above the maximum

combined moisture of extinction used in SPITFIRE of 30 %, on an oven-dry mass basis, when calculated for realistic live fuel moistures and fuel beds that contain a large live fuel component (note that e.g. in NFDRS 2016, described, e.g., in Andrews (2018), and Scott and Burgan (2005) 30 % is the minimum allowed live fuel moisture). However, as shown in Figure 2, despite this tendency of the parametrization to produce high combined fuel moistures when given realistic live grass moistures, SPITFIRE simulates an excessive amount of fire in grasslands. To examine this apparent contradiction, we implement a live

grass moisture output in LPJmL4-SPITFIRE, and subsequent versions. We find that the live grass moisture is unrealistically low, often to an extent that it does not comport with the conditions required for grasses to survive.

Live grass moistures above 100 % on a dry mass basis are often observed in the literature (e.g. Mendiguren et al., 2015; Brown et al., 2022; Yebra et al., 2019). Because these values occur frequently and are a natural part of the dynamics of grass phenology, a successful live grass moisture parametrization must be able to reproduce this. By its construction, the SPITFIRE

live grass moisture parametrization cannot exceed 100 %, as visible in Figure 5 a). Further, there are many regions, particularly on the Iberian Peninsula where the live grass moisture *on average* is below 10 %. In detail, the live grass moisture in SPITFIRE is given by:

$$M_{\text{f,lg}} = \max(0, \frac{10}{9}M_{\text{s},1} - \frac{1}{9}),$$

(2)

(Equation B2 in Thonicke et al., 2010), where $M_{\text{s},1}$ is the moisture content of the top soil layer relative to saturation (note

that for consistency in our variables we adopt the variable naming from Andrews (2018) rather than their forms in Thonicke et al. (2010)). The use of a soil moisture that is relative to its saturation point in Equation 2, which has a maximum value of 100 % by definition, results in a maximum live grass moisture of 100 %. In addition, as shown in Figure S2 in the supplement, the simulated live grass moisture is even lower when considering only months in which modelled fire occurs, illustrating the extremely low live grass moisture values that are required to permit fire spread in SPITFIRE.

Panels b) and c) of Figure 5 show the mean live grass moisture during the 3 winter months of December to February and the three summer months of June to August from 2003 to 2016. Because the live grass moisture is entirely dependent on the soil



moisture in the current parametrization, dormancy of grasses during the winter has no impact on their moisture content, and the live grass moisture during the winter, panel b), is, incorrectly, higher than in the summer, panel c), for most of the temperate and northern regions in Europe (see e.g., Bristiel et al., 2018; Keep et al., 2021; Sjöström and Granström, 2023). In the LPJ-GUESS

implementation of SPITFIRE, the issue of these seasonality dynamics was improved upon by treating phenologically inactive grasses as dead fuel, assigning to them the dead fuel moisture.

Finally, the live grass moisture of extinction, $M_\mathrm{x}$, of 20 % used in SPITFIRE is substantially lower than many realistic live grass fuel moistures. In Albini (1976), for example, the live fuel moisture of extinction is set to, at minimum, the dead fuel moisture of extinction (which in SPITFIRE is 30 %). Therefore, the value of 20 % is not supported by previous literature. In

the subsequent section we introduce new parametrizations.

**Improvements to the treatment of live grass moisture in SPITFIRE**

The issue of high live grass moistures resulting in excessive fire extinction is solved through the corrected weighting scheme in the implementation of the Rothermel equation. Specifically, Equation A11 is replaced with a separate treatment of live and dead fuel moistures, allowing for fire at higher live grass moisture contents. In addition to this, we have implemented the live

fuel moisture of extinction parametrization from Albini (1976) to replace the previous, excessively low, live grass moisture of extinction. These corrections now allow for more accurate, higher, live grass moistures to be implemented into the model.

In addition to the more accurate live grass moistures described below, we introduce a frequently-used curing function to account for transitions of grasses between dormant and active states. This curing function, originally developed for the 1978 version of the US National Fire Danger Rating system and published in Burgan (1979), and also described in detail in Scott

and Burgan (2005), transfers grass fuel loads from the living fuel category to an additional dead fuel category depending on its moisture content. Grasses that contain over 120 % water relative to their oven-dry mass are considered fully green, and are fully treated as live fuels, and grasses that contain less than 30 % water are considered fully cured and their load is transferred entirely to a dead fuel class. In between these endpoints, the proportion of fuel transferred is a linear function of the live grass moisture. The proportion of the live grasses that is transferred to a dead fuel class is given the same moisture content as the

finest, 1 h, dead fuel class and the moisture of extinction of the dead fuels.

This treatment of live grass moisture, for example, can represent conditions in which grasses are dormant in winter, and therefore have a moisture content that is passively determined by weather conditions, followed by a greenup period in spring. This greenup period is an important factor for fire behaviour in many grass-dominated regions, because the onset of warmer temperatures does not immediately translate to to green grasses, resulting in a well-aerated, dry, fine fuel-dominated litter bed

that is prone to high rates of fire spread (see, e.g., Sjöström and Granström, 2023; Burgan, 1979). Note that in the subsequent discussions of live grass moisture parametrizations and in Figure 5, the live grass moisture refers only to the moisture content of the proportion of the grasses that is not cured. Therefore, in regions, such as temperate Europe in winter, with a live grass moisture content of 30 %, i.e. fully cured, the moisture content of the fuel is purely determined by the dead fuel moisture, and the live grass moisture is, therefore, not representative of the overall fuel moisture. Similar considerations apply when there are

partially cured fuel beds in extremely dry regions. As an extreme example, a live grass moisture of 50 % on a day where the



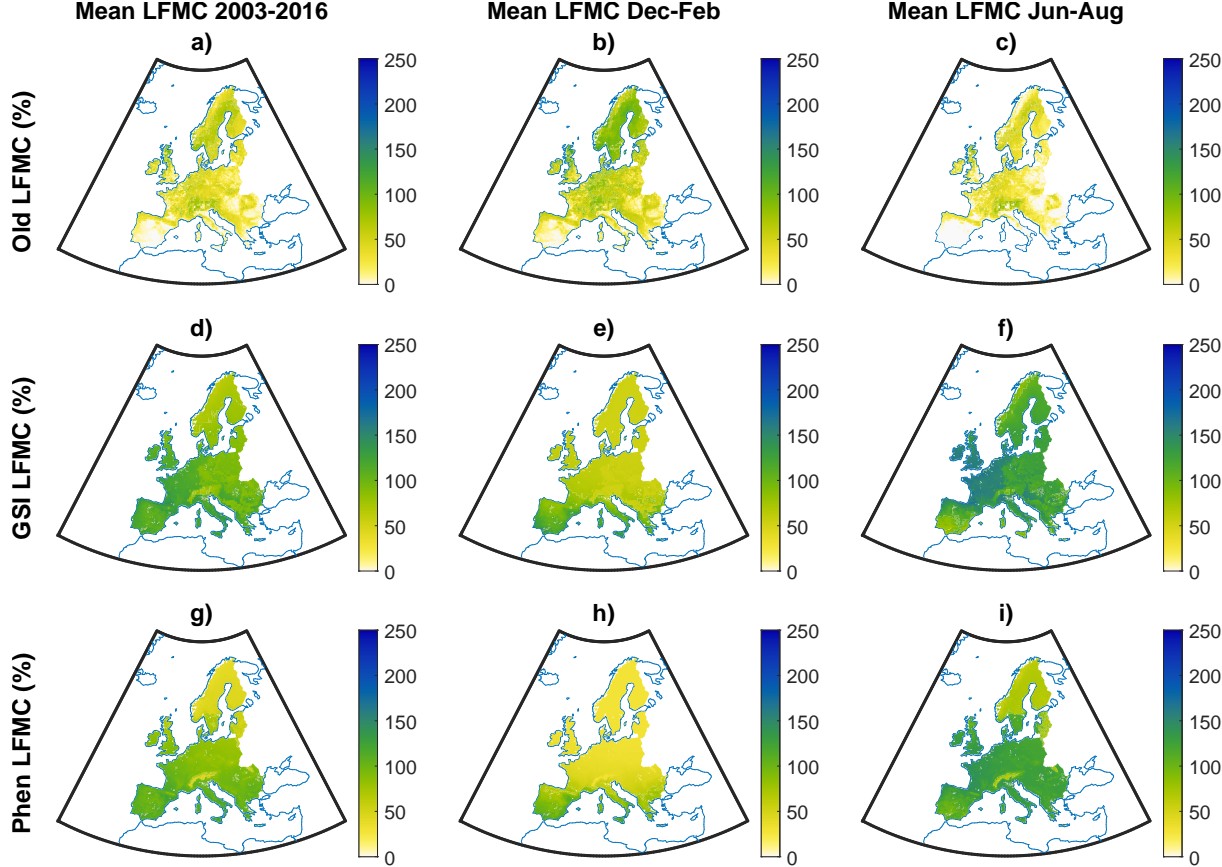

**Figure 5.** Original and updated live grass moisture parametrizations in LPJmL5.7-SPITFIRE on a 0.07° resolution in the European domain. Panels a) through c) show the original SPITFIRE parametrization from Thonicke et al. (2010), in % of liquid mass to oven-dry mass. Panels d) through f) show a growing season index-based parametrization for the live fuel moisture following NFDRS2016. Panels g) through i) show a newly developed live grass moisture parametrization based on LPJmL phenology. The columns show, from left to right, the mean live grass moisture from 2003-2016, the mean moisture content during the winter months of December to February, and the mean moisture content during the summer months of June to August, during the same time period. The two new livegrass moisture parametrizations show general agreement on average and during the months in which there are modelled fires. Note that the live grass moisture applies only to the component of the grasses that is not cured. Therefore, depending on the extent of curing and the dead fuel moisture, the overall moisture content of the grasses may be substantially lower.

fine dead fuels are completely dry (this may occur due to the much faster response of dead fine fuels to local weather conditions compared to live grasses), corresponds to an overall moisture content of the cured and not cured grass components of 11 %. We include these examples to help clarify the subsequent results but it should be noted that, because of the separate treatment of dead and live fuels, this overall moisture content is not directly used in the fire spread calculations and it is possible for the dead fuel component to burn while the live fuels do not ignite.




To calculate these new, more accurate live grass moistures, we introduce two updated approaches. The first of these is a
simple adoption of the herbaceous live fuel moisture parametrization from NFDRS2016, the US National Fire Danger Rating
System (described in, e.g., Andrews, 2018). Because this parametrization is dependent only on weather data and not the output
of the DGVM to which SPITFIRE is coupled, it can be easily ported to models other than LPJmL, and is not affected by issues
in modelled DGVM phenology. This parametrization is based on the Growing Season Index (GSI), developed by Jolly et al.
(2005), and calculates a vegetation phenological status by combining three functions: one of temperature, one of photoperiod,
and one of vapour pressure deficit. Each of these functions has a range from zero to one and these functions are multiplied
together so that each individual equation can limit the phenological status of vegetation. This combination is shown in Equation
3:

$$\mathrm{iGSI} = \mathrm{iVPD} \times \mathrm{iphoto} \times \mathrm{itemp}. \tag{3}$$

Here, iGSI refers to the daily growing season index. A 21-day running average of the iGSI is calculated to arrive at the
GSI and this is converted into a live grass moisture parametrization by setting a threshold value of 0.5 below which the grass
is dormant and using a linear relationship with endpoints of 30 % live grass moisture at a GSI of 0.5 and 250 % live grass
moisture at a GSI of 1. As noted in Krueger et al. (2022), this parametrization, although in use in the US, has not yet been
thoroughly tested. Therefore, we also introduce a new live grass moisture parametrization that can be used for European runs
of the SPITFIRE model.

As stated in Krueger et al. (2022), and further supported by other work, including Brown et al. (2022), soil moisture plays
an important role in shaping fire dynamics. One effect, of particular relevance here, is that the soil moisture is an important
parameter in determining herbaceous live fuel moisture. The GSI, in essence, includes this effect by proxy, via the VPD
function. The LPJmL DGVM has existing phenology functions that act in a similar manner to the GSI but use the soil moisture
calculated in the model rather than the VPD (an approach that was also suggested as a potential adjustment for modelled
phenology by Jolly et al. (2005)). We develop a new live grass moisture parametrization by making use of the phenology
functions of the grass PFTs in LPJmL, giving each function a greenup threshold below which the grass is considered dormant,
and fitting a linear function of the grass phenology to live grass moistures reported by Forkel et al. (2023). These functions are
shown in Figure 6, and the equation for livegrass moisture is:

$$M_{\mathrm{f,lg}} = k_{\mathrm{lg}} \times \mathrm{phen}, \tag{4}$$

subject to the condition that $30 \leq M_{\mathrm{f,lg}} \leq 250$, which we adopt from the GSI approach. Here, $k_{\mathrm{lg}}$ is an empirically-derived
scaling factor and $\mathrm{phen}$ is the phenological status of the grass as determined in LPJmL (a description of how phenology is
calculated in LPJmL can be found in Forkel et al., 2014). This phenological status is a function of mean daily air temperature,
total daily shortwave radiation, and modelled daily soil water content. The scaling factor $k_{\mathrm{lg}}$ is determined by defining a point
in the empirical data, i.e. a greenup threshold, where livegrass moisture begins to increase with phenology.

To ensure that the observed live grass moistures used to arrive at this parametrization are not overly impacted by other live
fuels, we select only grid cells that contain over 50 % cover by a specific grass PFT, and over 75 % natural vegetation, to fit these



functions. This also has the benefit of focusing on grid cells in which the Forkel et al. (2023) dataset has been shown to be most

accurate, i.e. non-forested areas. The shape of these derived functions, as seen in Figure 6, agrees well with data for temperate grasses, which is the herbaceous vegetation type present in the majority of Europe in the model ($R^2$ of 0.663). The function for polar grasses shows less strong agreement, with an $R^2$ of 0.29, perhaps due to the observed live fuel moisture being larger than 30 % for many points before the greenup threshold. Future parametrizations may experiment with different live grass moisture contents at the beginning of greenup. In general, however, the observed data agree well with the conceptual framework for the

GSI-based live grass moisture from NFDRS2016 that the live grass moisture increases linearly with increasing phenological status after a greenup threshold is reached.

The combined live grass moisture for a grid cell is then calculated using a mass-weighted average of the moisture content for each herbaceous PFT. For the current work, we limit this new parametrization to the European domain, reserving a global parametrization for future work. The live grass moistures produced when these parametrizations are integrated into the

vegetation model are shown in the second and third rows, panels d) through i), of Figure 5. Both parametrizations produce live grass moisture values that reflect the seasonal cycle of live grass moisture content for temperate and Nordic regions in Europe in which the grass is dormant in winter and active in summer. The LPJmL phenology-based live grass moisture shows a lower minimum in this cycle with generally dryer grasses in winter than the GSI-based parametrization. The GSI-based parametrization shows a slight divide between eastern and western Europe in the live grass moisture that is not present in the

phenology-based LFMC. This may be due to differences in the VPD included in the GSI parametrization that do not carry over into the soil moisture, and are therefore not present in the LPJmL phenology-based parametrization (for this reason this divide is also not seen in the old live grass moisture parametrization).

While the new live grass moisture parametrizations are brought more into agreement with established methods and the newly introduced LPJmL phenology-based live grass moisture shows a reasonably strong fit to satellite observations, some spatial

incongruencies with observed data remain. In particular, the live grass moisture in the summer in Mediterranean regions may be too high. As shown in the observations of Chuvieco et al. (2009) and Mendiguren et al. (2015), and examined experimentally for select species by Bristiel et al. (2018) and Keep et al. (2021), Mediterranean grasses often enter dormancy in the summer. This would correspond, in the case of dormancy of all grasses in a grid cell, to a live grass moisture of 30 %, i.e. the minimum value, and, through the curing function, all of the grass would be transferred to the dead fuel category. That this is not the case

in the modelled values may be for two reasons:

The first reason may be the parametrization of the grass PFTs in the LPJmL vegetation model version used for these results. In the model, the Temperate Herbaceous PFT covers much of Europe and, since there is only one phenology function for each PFT, albeit with different inputs given grid cell conditions, temperate grasses across Europe follow the same response to stresses. Because of this, differences in adaptation to drought stress may not be captured, and modelled grasses in the

Mediterranean may exhibit tendencies to remain active under hot and try conditions, as were observed for individuals from further northern regions by Bristiel et al. (2018) and Keep et al. (2021). Therefore, the excessive activity during dry periods may be a result of the manner in which vegetation is divided into PFTs by LPJmL not allowing for the differences at an intra-specific level observed by Bristiel et al. (2018) and Keep et al. (2021) to manifest in the modelled results. This topic of the



division of PFTs and its impact on model results is discussed further in Appendix C3. The issue of differences in adaptation,
of course, applies to the GSI-based grass moisture as well, which is currently based on a single phenology (although it may be
possible to adjust the thresholds used in the GSI equations on a regional basis in future).

The second reason for the high modelled grass moistures in Mediterranean summers may be due to over-estimations of
live grass fuel moisture content in remotely sensed data under very dry conditions. Our modelled values, while exceeding
those observed by Mendiguren et al. (2015) on the ground, are in much better agreement with values that they calculate based
on satellite observations of vegetation greenness, and that only depart substantially from the on-the-ground values during the
summer months. This potentially suggests a general challenge in deriving grass moisture from remotely sensed data, and future
versions of the live grass moisture parametrization in SPITFIRE may be further improved by also including local observations
(e.g., those in Yebra et al., 2019).

These uncertainties in the live grass moisture parametrization are mitigated somewhat by the fact that there are years, as
shown in Figure 2 of Chuvieco et al. (2009), in which Mediterranean grasses may remain active. These years are exceptions
rather than the rule, but they indicate that the modelled error is on the level of raising the frequency of an infrequent occurrence,
rather than creating entirely unrealistic conditions. In addition to this, the separate treatment of live and dead fuels, and the
new live grass moisture of extinction, which is generally higher than the dead fuel moisture of extinction, result in live grasses
that can still burn at higher moisture contents, and that fires can continue to burn in dead fuels even if this higher moisture
of extinction in live fuels is exceeded. Together with the curing function, which transfers a portion of grass to the dead cate-
gory beginning at 120 %, meaning that Mediterranean grasses in the summertime are generally treated as partially cured, this
substantially reduces the impact of the too high grass moistures, as shown by the model output in Sect. 3.3, particularly when
compared to the previous parametrization.

Due to the mitigating factors above, the generally good agreement with satellite observations shown in 6, and the substantial
improvement over the original live grass moisture parametrization in SPITFIRE, the updated live grass moistures shown here
represent a strong development in the context of SPITFIRE, and in including soil moistures in live grass moisture calculations.
Because further refinements of these parametrizations likely require changes on the DGVM side as well, they are outside of the
scope of the current work, in which our developments focus on the major issues in the existing SPITFIRE parametrizations.

We discuss the remaining, lesser sources of uncertainty, in the SPITFIRE model in the order of the model structure in Figure
1.

### 3.2.3 Fire danger index

As mentioned previously, there are 3 versions of the FDI implemented in different versions of SPITFIRE. The first, original
formulation given in Thonicke et al. (2010) is different from the formulation in Venevsky et al. (2002) upon which it is based.
While Venevsky et al. (2002) use an exponential function of the Nesterov index together with a tuning parameter to arrive at
the FDI directly, Thonicke et al. (2010) apply the Nesterov Index to calculate the dead fuel moisture as an intermediate step
to calculating the FDI (Equation 6). To our knowledge, this equation remains untested against observed fuel moistures and an
examination of its ability to predict dead fuel moisture may be relevant to versions of SPITFIRE that depend on it. We apply




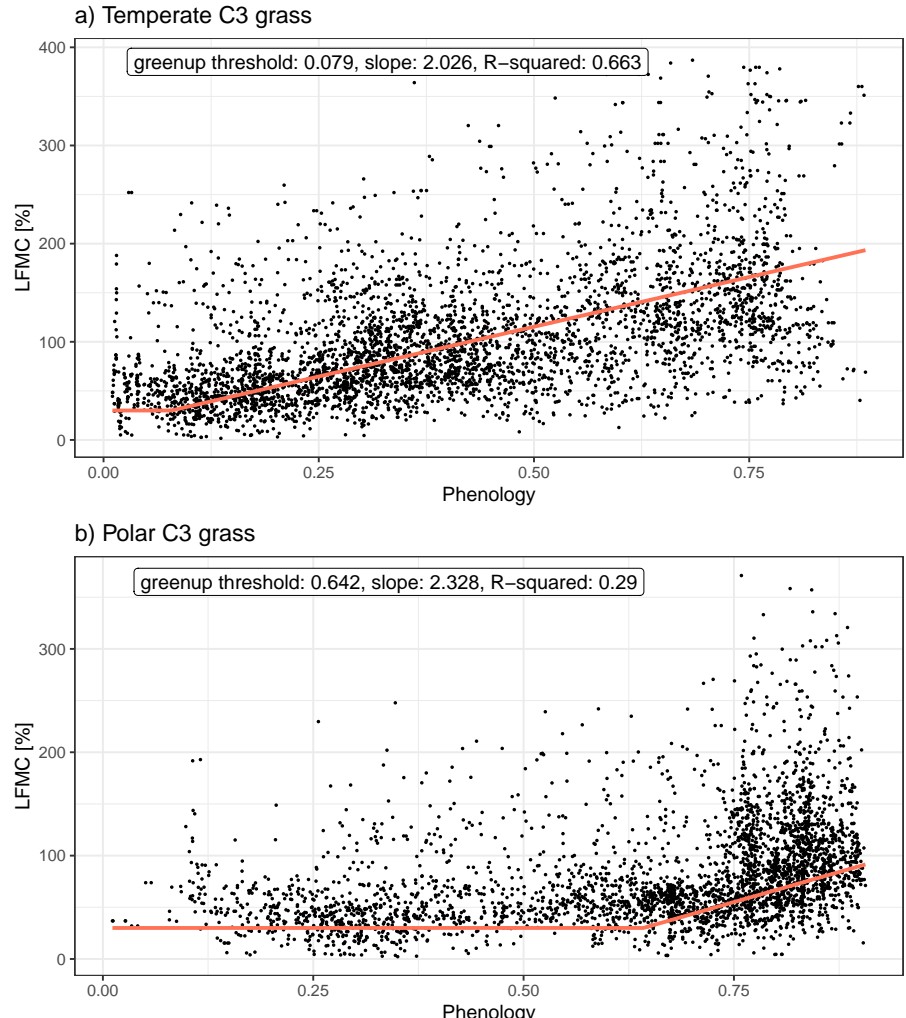

**Figure 6.** New PFT-specific live grass moisture functions in LPJmL-SPITFIRE. Here, phenology refers to the numerical phenological status of the grasses, i.e. how active they are, as calculated in LPJmL using the parametrization from Forkel et al. (2014). Points indicate measured values.

the VPD-based FDI from Drüke et al. (2019) to rectify this and introduce a new dead fuel moisture parametrization, described in detail in the fire spread component, Sect. 3.2.5.



### 3.2.4 Fire duration function

The fire duration function in SPITFIRE is shown for reference in Figure S3 in the supplement, and given by Equation 14 in Thonicke et al. (2010):

$$t_{\text{fire}} = \frac{241}{1 + 240e^{-11.06 \times \text{FDI}}}. \tag{5}$$

This function has a maximum value of 241 minutes and the functional form has a step-like nature such that at low FDI values the fire durations are extremely low, followed by a sharp increase around FDI values of 0.5. This results in fires with durations that are often substantially below the 241 minute maximum. In practice, fires in the LPJmL4-SPITFIRE model have a median duration of about 30 minutes. In addition, the SPITFIRE model does not allow for fires that continue burning over several days, despite this being a common phenomenon (see e.g.Andela et al. (2019) where the median fire duration globally is 3 days). Dividing multi-day fires into separate single day fires suffers from the issue that the elliptical fire spread in SPITFIRE results in a quadratic dependence of fire size on fire duration. Therefore, e.g., two single day fires have a combined fire size only half as large as one two day fire, all else being equal. This substantial downward bias in fire size is a large compensating error that prevents the strong upward biases, due to the incorrect weighting factors in the fire spread component of the model, from causing highly overestimated modelled burnt areas. Now that these upward biases have been corrected, however, it is possible to implement a more realistic fire duration function.

For this purpose, we have introduced a multi-day burning algorithm into SPITFIRE. This allows fires that begin burning on one day to continue into the next, with an adjustable maximum number of days. We have currently implemented an FDI-based condition for stopping the spread of these multi-day fires, where all fires in a given grid cell are extinguished if the FDI reaches a value below 0.005. This condition operates in addition to the previous condition in SPITFIRE that fires are extinguished if the calculated fireline intensity is below a specified threshold. These conditions, in particular the FDI threshold, are set as initial placeholders and further work is required to develop a fire extinction function that can properly capture the conditions under which fires no longer spread. One benefit of the multi-day fire approach we introduce is that it allows for the existence of fires that experience less fire spread on some days, e.g. due to lower wind speeds, but continue burning and experience more spread on subsequent days.

The principle mechanism underlying the multi-day fire algorithm is that the major axis of the elliptical fires in SPITFIRE grows according to the daily wind speed and fuel bed parameters. The length-to-breadth ratio is calculated using the mean wind speed over the full set of days that the fire spreads, assuming that the fire spread continues broadly in the same direction each day. These simplifications are necessary based on the current model inputs and the state of knowledge of fire spread at the SPITFIRE scales. The fundamental challenge is that including changes in spread direction would require the use of coarse grid cell averaged wind directions in addition to the current grid cell averaged wind speeds. These directions can be quite variable on the smaller scales at which a fire spreads and, therefore, may not be sufficiently represented by the grid cell average. Because of these simplifications, the algorithm we have introduced should be considered experimental, and requires further development and research to apply fully. However, it represents a pathway for removing the bias caused by including only single-day fires.





In addition to the multi-day algorithm, we have implemented the possibility of setting a daily maximum fire duration more than 4 hours in the model, allowing fires to burn longer on a given day. However, the step-like shape of the function, and the fact that modelled FDI values are generally substantially lower than the location of the step in the function, results in daily fire durations that are often less than 30 minutes. As a preliminary solution, we have also introduced the possibility of a user-specified minimum fire duration into the equation and replaced the factor of -11.06, which was set so that an FDI of 0.5 corresponds to 1/2 of the maximum fire duration of 241 minutes, with an adjustable parameter. The new equation is:

$$t_{\text{fire}} = \frac{t_{\text{fire,max}} + 1}{1 + (t_{\text{fire,max}}/t_{\text{fire,min}} - 1)e^{k_{\text{fireduration}} \times \text{FDI}}}.$$ (6)

Where, $t_{\text{fire,max}}$ and $t_{\text{fire,min}}$ are the maximum and minimum fire durations respectively, in minutes, and $k_{\text{fireduration}}$ is a tuning parameter that controls the slope of the function before it saturates towards the maximum duration. A plot of this function is shown in Figure S4 in the supplement.

These updates improve upon a conceptual limitation in SPITFIRE, and establish a technical framework in the model code for calculating multi-day fire spread. Therefore, they act as a strengthened platform upon which to build more detailed fire duration functions. These functions are written so that, if desired by the user, the previous SPITFIRE fire duration function can be simply recovered by adjusting options in the model.

### 3.2.5 Fire spread

To resolve the uncertainty regarding the Nesterov-based fuel moisture content described in the FDI section above, we replace the fuel moisture parametrization with the new parametrization developed for LPJmL by Lutz et al. (2019). This parametrization contains a full water balance for the litter that includes interception of precipitation by the litter, infiltration and runoff from the litter, and a temperature-based evaporation function. In addition to providing a stronger theoretical basis for the dead fuel moisture content, this has the benefit of creating greater internal consistency in LPJmL-SPITFIRE. Further improvements to the dead fuel moisture content component of the model may be made in future by distinguishing between the moisture content of different fuel size classes using, e.g., the Nelson dead fuel moisture model (Nelson, 2000; Carlson et al., 2007). We have made provisions for this in the model code by treating the moisture content of the different fuel classes separately, giving each the current uniform value. Values from a future parametrization can therefore simply be input into the model.

In addition, we discovered an LPJmL-SPITFIRE-specific bug in the fuel load of live grass, in which the amount of live grass was scaled by the phenology to represent curing. However, the cured component of the grass was then not accounted for. Our introduction of the curing function has resolved this issue.

Another LPJmL-SPITFIRE-specific bug was fixed in which the total amount of burnable live grass is overwritten during a loop over each PFT. The former version resulted in the total amount of consumable live grass being equal to the amount given by the final PFT. This reduction may account in part for why the extremely severe fires caused by the biases discussed above did not result in non-physically high fire carbon emissions.

Finally, to avoid excessive burning on open grasslands, we adopt the wind limitation function previously applied to JSBACH-SPITFIRE by Lasslop et al. (2014) in LPJmL-SPITFIRE. This function is a version of the original wind limit from Rothermel



(1972), updated by Andrews et al. (2013). We also include the original wind limit function as an alternative option. In our tests, due to the relatively low values of grid cell and daily averaged wind speeds, this wind limit is often not reached, but we introduce it to provide for future parametrizations that may include higher wind speeds.

### 3.2.6 Mortality

We identified several potential improvements in the cambial damage mortality function in SPITFIRE. The cambial damage function is largely based on the work of Peterson and Ryan (1986), with a few modifications. In contrast to Peterson and Ryan (1986), who use a weighted averaging scheme to arrive at the residence time, the residence time in SPITFIRE is calculated using a simple ratio of the amount of fuel consumed to Rothermel's reaction velocity, $\Gamma$. In this ratio, the amount of fuel consumed is simply calculated using an average of the amount of 1-, 10-, and 100h fuel consumed, i.e. a sum of the three

values divided by three. In Peterson and Ryan (1986) the fuel consumption of different fuel classes is combined into a common value using a weighting based on the amount of fuel bed area that is covered by the individual components. To address this, we simplify the residence time equation to the simple function of fuel surface-area-to-volume ratio given by Albini (1976). The surface-area-to-volume ratio input into this parametrization results from the Rothermel weighting scheme described previously, and therefore accounts for fuel bed heterogeneity in a more conceptually sound manner.

In the subsequent step of the cambial mortality function, the amount of time that the vegetation experiences lethal heat $\tau_L$ is calculated from the residence time. In Peterson and Ryan (1986) this value is 5 times the residence time, whereas in SPITFIRE this value is only 2 times the residence time. As there is no conceptual reason for this difference, we reinstate the value of 5. This substantially lower residence time may have also compensated somewhat for the excessive fire intensity caused by the biases in the application of the Rothermel equation.

The equation for the probability of mortality due to cambial damage that is based on this burning time is given in Equation 19 of Thonicke et al. (2010). We discovered some disagreement between this equation and the cited literature, Peterson and Ryan (1986), on which it is based. Specifically. Peterson and Ryan (1986) combine cambial damage and crown scorch into a single equation, in their Equation 11, whereas SPITFIRE treats the two separately and combines the probabilities of mortality due to cambial damage and crown scorch under the assumption that these are independent. This assumption does not necessarily

agree with the physical processes of mortality wherein, e.g., a more intense fire produces more heat as well as taller flames, and is therefore more likely to both damage the cambium and scorch the crown of a tree. We therefore implement the Peterson and Ryan (1986) probability of mortality calculation as a simplified parametrization that removes the uncertainty associated with the theoretical basis of the SPITFIRE functions. This also allows for an easier interpretation of model results and forms a basis for future work improving the mortality parametrization. The equation is given by:

$$P_m = c_{\mathrm{k}}^{\tau_c/\tau_{\mathrm{L}}-0.5}. \tag{7}$$

Where $c_{\mathrm{k}}$ is the fraction of the crown that is scorched, determined using the scorch height and tree geometry, $\tau_c$ is the critical time for cambial damage, in minutes, and $\tau_{\mathrm{L}}$ is the amount of time, in minutes, that a tree experiences lethal heat.





Finally, the equation for the $c_k$ parameter, the parameter describing the fraction of a tree crown that is killed by fire, in Peterson and Ryan (1986) is based on a paraboloid crown structure. In SPITFIRE this was replaced with a cylindrical crown

structure, reducing the amount of the crown that is near the ground (this change is also present in the adaptation of the $c_k$ parameter for the crown fire parametrization in Ward et al. (2018)). Because the new fire spread formulation avoids excessive crown scorch due to high-intensity fires, we return to the formulation of Peterson and Ryan (1986) for the sake of conceptual uniformity and the more realistic crown structure described therein.

### 3.3    Updated model version at the European scale

To examine what impact the changes from the previous sections have once incorporated into the model, we have created a preliminary model version specifically for the European domain. We choose this area as a test case because we aim to restrict the amount of variability the model has to account for on a global scale and due to the involvement of the SPITFIRE model in the FirEUrisk project (https://fireurisk.eu). The new model version was developed using data available through the FireEUrisk project on a $0.07°$ scale, also allowing for less sub-grid variability than the usual $0.5°$ scale. The results shown here are

preliminary in nature and intended to ensure that the changes made to the model perform reasonably. A full new version of the SPITFIRE model is reserved for further work pending additional testing and operation on a global scale.

In addition to implementing the changes listed above, we have conducted a thorough code cleanup and have introduced a new changelog in the model. We designate the current model version as LPJmL-SPITFIRE1.9, reserving the label LPJmL-SPITFIRE2.0 for a version that has been tested at the global scale. For the new model version we have re-tuned the parameters

for the European domain. We follow the standard tuning approach for the model and use the PFT-specific $\alpha$ parameters as the main tuning parameter, but also tune various ignition and fire duration parameters listed below. The values of these parameters for each PFT are shown in Table 1. We set the fire duration to a minimum of 2 hours and a maximum of 7 hours. The maximum number of consecutive days that a fire can burn is set to 3. On managed grasslands, which forms a small portion of the modelled grid cells, these parameters are set to a minimum of 1 minute, a maximum of 2 hours, and a maximum of 1 day respectively.

In the fire duration function, Equation 6, the factor, $k_{\text{fireduration}}$, multiplying the FDI is set to -8 from the previous -11.06, allowing a more gradual rise in fire duration with increasing FDI. Finally, in the human ignitions function, Equation B1, the coefficient at the start of the equation, which we now call $k_{\text{ignitions}}$, was changed from 30 to 165. We have applied a fairly liberal tuning to these parameters as there is no clearly established value for them currently, and our current aim is largely to compare the impact of the changes we have introduced.

We compared the results of the new model version with the standard SPITFIRE model (Figure 7). For this comparison we implement both model versions in LPJmL5.7 and therefore operate with its included litter moisture. To allow for a direct comparison of the fire spread and mortality processes, we apply the new tuning parameters and the multi-day fire algorithm to the old model version as well. In all other respects the new model version contains the improvements we have mentioned and the old model version does not. The results for the old model version implemented in LPJmL5.7 show the same biases as

the version implemented in LPJmL4 (shown previously in Figure 2). Namely, there is an extreme overburning in grassland-dominated grid cells, illustrated particularly in panel f), despite forested grid cells showing rough agreement for many years.





**Table 1.** Tuning parameters for the European model version, $\alpha$ parameters are the FDI scaling parameters for each PFT, $t_{\text{fire,min}}$ and $t_{\text{fire,max}}$ are the minimum and maximum daily fire durations in minutes, $k_{\text{fireduration}}$ is the slope factor applied to the daily fire duration, $k_{\text{ignitions}}$ is the scaling factor applied to human ignitions, and fire days is the maximum number of days for which fires are allowed to burn

| $\alpha_{\text{TrH}}$ | $\alpha_{\text{TH}}$ | $\alpha_{\text{PH}}$ | $\alpha_{\text{TBE}}$ | $\alpha_{\text{TNE}}$ | $\alpha_{\text{TBS}}$ | $\alpha_{\text{BNE}}$ | $\alpha_{\text{BNS}}$ | $\alpha_{\text{BBS}}$ | $t_{\text{fire,min}}$ | $t_{\text{fire,max}}$ | $k_{\text{fireduration}}$ | $k_{\text{ignitions}}$ | fire days |
|---|---|---|---|---|---|---|---|---|---|---|---|---|---|
| 4 | 4 | 7 | 10 | 10 | 10 | 15 | 15 | 15 | 120 | 480 | -8 | 165 | 3 |

The new model version shows a substantial improvement in this regard, showing better agreement in grass-dominated grid cells and reasonable agreement in tree-dominated grid cells. This improvement is visible in central areas of the Iberian peninsula, for example, where modelled grasslands (see Figure S5) result in higher burnt area in the old model version than the new. The improvement in this division is particularly noteworthy since this division was not used as a target when tuning the model, with only the broad spatial pattern and annual burnt area being targets. The new model version also results in a much lower average rate of spread, shown most clearly in panel i), and a reduction in the amount of trees that undergo fire mortality, shown in figure S5 in the supplement. Therefore, the upward biases shown in Figure 4 had a substantial impact on model results that is now reduced. Note that due to the lack of a prescribed firestarts input at the $0.07°$ scale, we do not perform the prescribed ignitions validation for this smaller scale version, and reserve such tests for future larger scale versions in which the fire duration function can be further developed as well.

Some issues remain in the new model version, however. Most importantly, the model's lack of ability to reproduce regions of high burnt area, such as Portugal, illustrates that there remain factors determining burnt area that the model currently does not capture, including fire suppression, fragmentation effects, and greater flammability of some species of vegetation. In addition, there is substantially too low burnt area in eastern Europe in the new model version. This may be due to an under representation of ignition factors in that region, as the rate of spread, shown in panel i), remains high in this area. A more detailed study of this region is out of the scope of this work, but future work may examine the modelling of these sources of ignition further.

The fixes to the implementation of the Rothermel equation result in substantial reductions in the rate of spread and the elimination of regions where there are extremely high rates of spread in the old model version. Due to the lack of reliable spread rates in satellite-based products, we reserve the testing of modelled rates of spread against local data for future work. Overall, the new model version shows a substantial improvement in the relative amount of fire in forested grid cells as opposed to those dominated by grassland, and has a substantially corrected theoretical basis. It can therefore be adopted as a foundation upon which to build future model versions.

## 3.4 Model status

We have identified several sources of uncertainty in the SPITFIRE model. The most important of these are the inaccurate implementation of the Rothermel equation and the unrealistically low live grass moistures. We resolve these issues in part by correcting the implementation of the Rothermel equation in SPITFIRE and by introducing new live grass moisture parametrizations. These parametrizations produce much more realistic live grass moistures. The applied LPJmL phenology-based parametriza-



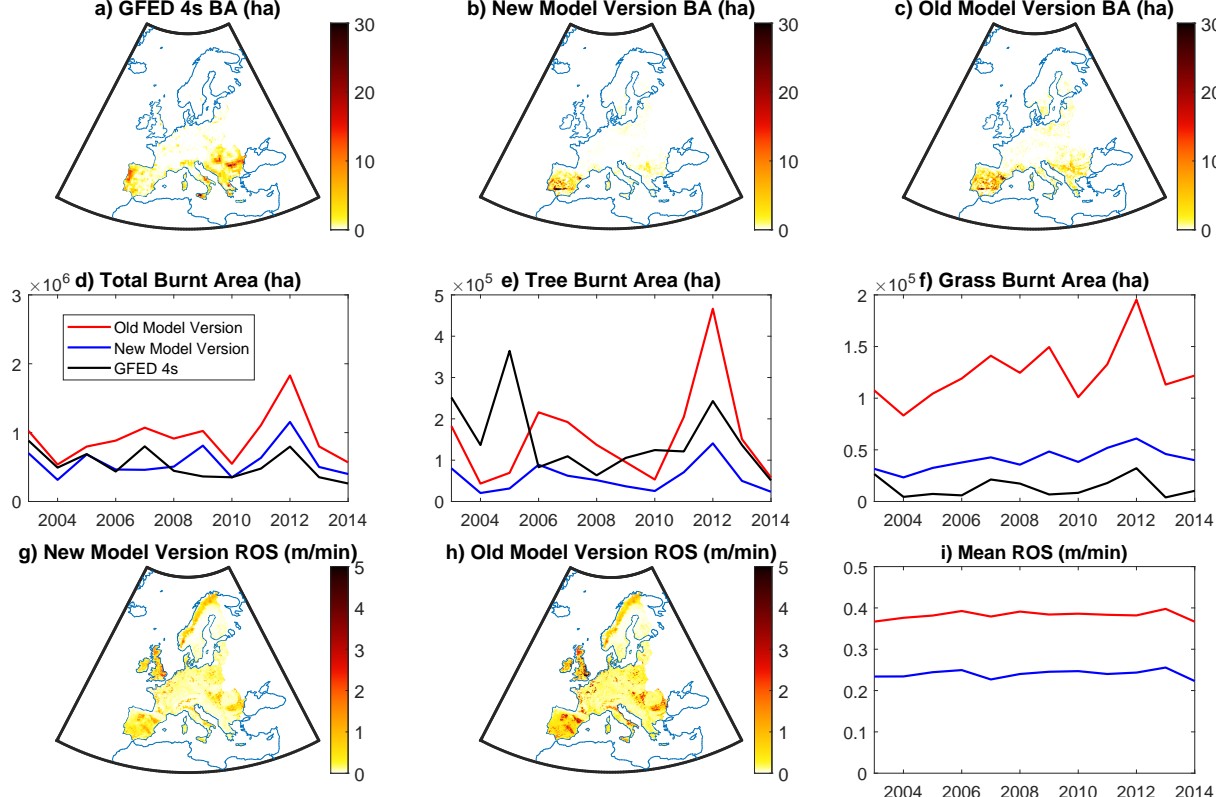

**Figure 7.** Impact of changes to the LPJmL5.7-SPITFIRE model in the European domain. Maps in panels a) through c) show mean burnt area per grid cell over the 2003-2014 simulation time period. Time series in panels d) through f) show total annual burnt area over the simulation domain. Tree burnt area and grass burnt area refer to burnt area in grid cells whose area is made up of over 50 % tree PFTs and Grass PFTs respectively. Panels g) through i) show maps and time series of the rate of spread calculated in the new and old model versions. The time series in this row contains the mean annual value rather than the annual sum as in the row above. The new model version shows a substantial improvement in the relative amounts of burnt area in forests compared to grasslands. It is also less volatile, with less extreme inter-annual variability.

tion, in particular, represents an innovative approach to incorporating soil moisture in the calculation of herbaceous live fuel moisture, an important development, the need for which has been identified in previous studies (Brown et al., 2022; Krueger et al., 2022; Jolly et al., 2005). Our test of the new SPITFIRE version in the European domain shows that it is capable of broadly representing burnt area, similar to the previous model version, while providing a substantially strengthened theoretical basis.

Because of the extensive changes we have made to the LPJmL-SPITFIRE model, and the uncertainties we have discovered, it is relevant at this point to discuss the current status of the model. In addition to summarizing the changes that we have made, we include a discussion of sources of uncertainty that should be taken into account when using the model and that may





be improved upon in future. While many of these sources of uncertainty are familiar to those experienced with fire-enabled DGVMs, we include a detailed discussion here both for the sake of transparency and to support future work. It is our aim to clarify these sources of uncertainty for those unfamiliar with the SPITFIRE model but who may be users of model results, are
new adopters of the model, or may have expertise in other areas of fire research that could be used to help develop improved model parametrizations. In many cases the improved model foundations that we have presented here enable these improvements to be made to a greater extent. For example, the ignitions parametrizations can now be improved using more realistic data as they no longer need to compensate for the upward biases in the fire spread component of the model. It should be noted that the impact of these sources of uncertainty is dependent on the application to which the model is put, and the specific use case
should be considered when interpreting their importance. The changes made to LPJmL-SPITFIRE and key areas that remain for future development are summarized in Figure 8, structured in the same manner as Figure 1, and discussed in detail below.

### 3.4.1 Uncertainties and areas for future development in specific model components

Areas for future development in specific model components are discussed in Appendix B. For the ignitions component, we highlight the uncertainty of the human ignitions parametrization and the temporal averaging of lightning ignitions that leads
to small fractions of an ignition each day that may not align with the timing of associated precipitation (Appendix B1). These components, in particular, have gained greater scope for future developments through the model improvements in this work, since they are no longer required to compensate for excessive fire spread. For the fire spread component, we highlight simplifications in the moisture of extinction and surface-area-to-volume ratios that may reduce model accuracy as well as aspects that could be important to fire spread that are currently not modelled (Appendix B2). A particular detail about the fire spread
component that emerged in our literature analysis, and where a new parametrization remains necessary is the calculation of fuel bulk density.

**Fuel bulk density**

A major outstanding issue in the fire spread component of the model is its treatment of the fuel bulk density, i.e. how tightly packed the fuel beds are, an important input in the Rothermel model. In its fire spread component, SPITFIRE uses literature-
based values for the fuel bulk density for each plant functional type. These values are then modified by the following equation given in table A1 in Thonicke et al. (2010):

$$\rho_b = \frac{1}{n} \sum_{\mathrm{PFT}=1}^{n} \rho_{b,\mathrm{PFT}}(w_{0,\mathrm{1h}} + 0.2w_{0,\mathrm{10h}} + w_{0,\mathrm{100h}}) \tag{8}$$

Here, $n$ is the number of plant functional types in a given grid cell, $\rho_{b,PFT}$ is the PFT-specific fuel bulk density value, in $\mathrm{kg\,m^{-3}}$, and $w_{0,x}$ is the fuel load in fuel class $x$, in $\mathrm{kg\,m^{-2}}$. The justification for this equation is based on a statement in
Brown (1981) that when the 10-hour fuel class is eliminated from calculations, the calculated bulk densities averaged 80 percent of their previous values. Equation 8 does not have this effect. Rather, it leads to a scaling of the bulk density depending on how much total fuel load is present. Therefore, this question of parametrizing fuel bulk density in an aggregated, grid cell-scale manner remains open, and a new parametrization is required.





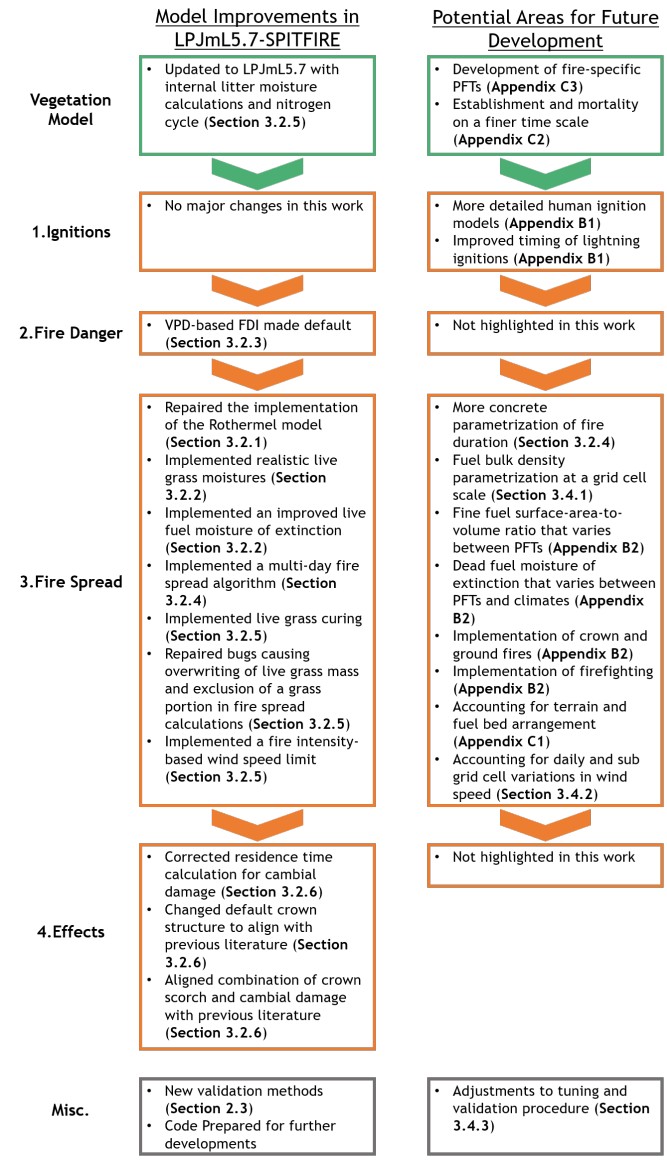

**Figure 8.** Summary of model changes implemented in LPJmL5.7-SPITFIRE and general areas for potential future development in SPIFIRE that were identified, structured in the same manner as Figure 1. The key changes discussed in this manuscript are the repaired implementation of the Rothermel model and the realistic live grass moistures (the first two bullet points in the fire spread component). The importance of the areas for future developments is dependent on the desired application of the model. Implementing these changes is made much more feasible by the model development in this work.





### 3.4.2 The impact of model resolution

In addition to the sources of uncertainty that we have highlighted above, there are several sources of uncertainty that arise due to the resolution of SPITFIRE and attached DGVMs. In several cases, which we highlight, these sources of uncertainty can be better mitigated due to the improved theoretical basis for SPITFIRE that we have established in this work.

Model resolution is one area where global fire models, necessarily, differ from predictive models used in an operational context such as FARSITE and FlamMap (Finney, 1998, 2006). There are three domains in which resolution plays an important

role in the SPITFIRE model. The first is the spatial resolution, i.e. the size of the grid cells, the second is the temporal resolution, and the third is the separation of global vegetation into PFTs. In all of these cases a certain coarseness is required to feasibly run the model on a global scale. We include a detailed discussion of these three domains as guidance for model users in Appendix C.

We particularly highlight the effects of the information given to SPITFIRE by associated DGVMs and cases where these

differ from one DGVM to another. Questions of vegetation size, arrangement, and what times during the year fuel beds are replenished are particularly relevant, as well as the choice of PFTs and how they align with fire characteristics for the regions under study. Spatially, challenges may be posed by calculating fire spread in fuel beds that are homogenized across a grid cell, and that therefore do not contain fragmentation effects or changes in fire behaviour between differently vegetated sub-grid regions (Appendix C1). Temporally, annual time steps of vegetation establishment and mortality may impact regions with

bimodal fire seasons where vegetation is replenished in the middle of the year, and regions where the fire season encompasses the turn of the year, resulting in unrealistic developments in modelled fuel beds (Appendix C2). A common issue in several DGVMs with which SPITFIRE operates is the lack of a shrub PFT for regions in which shrubs are important parts of the fire regime, and we have made provisions in our updated fire spread parametrizations to facilitate the inclusion of shrubs in future (Appendix C3). As a representative example, and one that is most relevant to the model improvements in this work, we

highlight the impact of model resolution on wind speeds here.

**Relevance for the incorporation of wind speeds**

An area where the model improvements presented in this work can contribute substantially to future developments is the wind speeds input into the Rothermel model. These are particularly subject to the impacts of model resolution. Spatially, interactions between terrain and wind, in particular, could be examined more closely, as complex terrain can have a substantial

effect on wind speeds local to the flame front in a manner that is not reflected by the grid cell average wind speed (see, e.g., the strong wind speed variance across domains with complex terrain shown by Jung and Schindler, 2020). Temporally, wind speeds are often lower at night than during the day (e.g., Ephrath et al., 1996; He et al., 2013). Therefore, the daily averaged wind speed currently used as an input to the model may not be reflective of the wind speed during the hours in which fires are most likely to spread. Future work could examine the feasibility of using averaged wind speeds over the hours during which the

VPD meets some threshold for fire spread, as discussed in the context of fire duration in Sect. 3.2.4, or, more simply, whether midday wind speeds may be more appropriate as an input parameter than the daily average. With the removal, in the new



SPITFIRE version, of the strong upward biases in rate of spread and scorch height caused by the previous implementation of the Rothermel model it may be possible to incorporate these higher wind speeds effectively.

### 3.4.3   Model tuning and validation

The fact that the issues we have highlighted, particularly the bias in the implementation of the Rothermel equation and the unreasonably low live grass moistures, were able to remain undetected in previous versions of SPITFIRE, suggests that more detailed validation procedures may be beneficial to future model versions. In this work, we developed two new validation methods, namely the division of burnt area by vegetation and the use of prescribed firestarts, that have shown an ability to identify underlying issues not revealed by standard validation procedures. This suggests that the proposed validation methods,

in addition to the more detailed methods already in use in the FireMIP project (Rabin et al., 2017), may be highly useful in future developments. An additional element that was highlighted by the results in this work is the potential ability of model tuning to obfuscate biases by creating errors that compensate for one another (as illustrated, for example, by the reasonable agreement of mapped burnt area in Figure 2 that did not arise from a reasonable balance of fire spread and ignitions as shown in Figure 3).

Model tuning is a key component of the development process for global models in general and process-based global fire models in particular (e.g., Hourdin et al., 2017; Hantson et al., 2016). Because compounding errors can often arise due to, e.g., measurement uncertainties in observed input parameters or modelling uncertainties in modelled input parameters, tuning fire model equations based on observed data is often necessary if the goal is to align model output with these data. In the case of the SPITFIRE model, process parametrizations are generally tuned to achieve agreement between modelled and satellite-based

mean burnt area maps, as well as annual time series of burnt area, over the modelled domain. Generally, the parameters used for model tuning are those which are the most uncertain due to a lack of data constraining their values. Most commonly, the number of human ignitions and the PFT-dependent scaling parameters applied to the fire danger index are the main levers by which model outputs are optimized. These parameters are adjusted by hand until the maps and time series in question achieve acceptable agreement with their observed counterparts. It should be noted as well that the DGVMs coupled to SPITFIRE

contain their own tuning processes that are similar to this approach albeit with different targets (e.g., von Bloh et al., 2018a; Scheiter et al., 2013). Therefore, the manner and targets of the tuning procedure may be a point of consideration for future users of SPITFIRE, particularly if values from particular sub-components of the model are desired outputs. For example, an application in which the number of fires is an important output, in addition to the burnt area, would benefit from a tuning procedure that takes into account the prescribed fire starts validation at an intermediate step.

More generally, and importantly for future predictions, the fact that the validation dataset is used to inform the model tuning approach in an iterative manner results in a substantial transfer of information from the former to the latter that weakens the model validation, as the validation dataset is no longer an independent source of information. In objective tuning methods, as discussed by Hourdin et al. (2017), and applied to the SPITFIRE model for a South American domain by Drüke et al. (2019), it is common practice to divide data into separate datasets for training and for testing, resulting in a reduced transfer of

information. The objective tuning approach is not a panacea, however, as it can result in tuning parameters that are not physi-





cally justified, and this must therefore be carefully monitored by modelers. However, if such unphysical parameters are arrived at, this can also be valuable information about the model, as it indicates model biases that may require reparameterization (Hourdin et al., 2017).

The division of validation data into training and test datasets could also be implemented for cross-validation of the hand tuning approach commonly used in SPITFIRE. The performance of SPITFIRE in grid cells that are not used in the tuning process would give a more robust estimate of model performance outside of its training data (see, e.g., the general discussion of cross-validation in Morin and Davis, 2017). The issue of increasing uncertainty outside of the training data may also be visible in the strong agreement of global fire models, in general, within the time period for which satellite data is available, but their strong divergence outside of that time period (e.g. Teckentrup et al., 2019). While such upgrades to the model tuning approach in LPJmL-SPITFIRE are out of the scope of the current manuscript, we have sought to discuss our model tuning approach transparently to allow for greater clarity in the interpretation of model results, and suggest the cross-validation approach here for future, full SPITFIRE versions.

### 3.4.4 Conditions for best performance of LPJmL-SPITFIRE1.9

Based on the qualitative and quantitative analysis in this work we can outline the conditions under which we anticipate the best model performance for model versions based on LPJmL-SPITFIRE1.9. Note that not meeting all of these conditions does not preclude satisfactory model performance, they simply indicate ideal conditions. Also, because of the possibility for model tuning to balance errors in one spatial region with errors in another, these conditions do not necessarily correspond to the accuracy of final model results in specific geographic regions. However, a focus on these regions may be effective when conducting model developments. The conditions for best model performance given the current status of the model are ones where:

- The coupled vegetation model accurately represents fuel beds.

- The species that dominate the fire regime align with the plant functional types of the DGVM (e.g. the fire regime is not dominated by shrubs in a DGVM that doesn't model them).

- Potential human ignitions occur steadily throughout the year (although successful ignitions may still vary considerably due to meteorological conditions), and lightning ignitions do not vary substantially between years, or are not a major ignition source.

- Fires are generally short in duration, and are therefore less subject to the uncertainties in propagating longer duration fires.

- The fire regime in a given location is dominated by surface fires, and fire suppression does not have a strong impact on overall burnt area.

- The region has a subhumid or humid climate, as defined by Scott and Burgan (2005), to ensure the dead fuel moisture of extinction in SPITFIRE is accurate.



– Vegetation in a grid cell is homogeneous, and evenly distributed.

– Terrain in a grid cell is flat.

– Fires occur toward the middle of the year in DGVMs that calculate annual establishment and mortality.

– Wind speeds are temporally and spatially homogeneous, and wind directions are steady.

## 4 Conclusions

We have undertaken a thorough review of the global fire model SPITFIRE to identify and better understand the sources of uncertainty in the model, and the cause of several known issues in the model results. We have found that two major sources of

error exist in the model. First, the model contains an incorrect implementation of the Rothermel fire spread model that can result in substantially too large and too intense fires. Second, the model contains a live grass moisture parametrization that results in unrealistically low moisture contents. We correct these issues with a corrected implementation of the Rothermel model and a novel live grass moisture parametrization for the European domain. Other sources of uncertainty are identified and partially corrected. This results in an updated version of the model that is more aligned with the physical basis of wildland fire spread

and allows for future additions of more accurate parametrizations to other parts of the model. The updated model retains the ability of previous model versions to broadly represent spatial patterns of burnt area. Results in the European domain show a reduction in excessive burning on modelled grasslands and reduced tree mortality. Further work is required to test the SPITFIRE model in the global and European domains and to address challenges of the modelling scale and required reparametrizations. By improving the theoretical basis of the SPITFIRE model and highlighting directions of future model development we have

sought to create a foundation upon which such work can be built. The new live grass moisture parametrization we introduce here, in particular, is an example of such a development, as the model no longer requires unrealistic live grass moistures for the fire spread component to function, allowing for a representation that better captures seasonal moisture dynamics.

*Code and data availability.* The current version of LPJmL is archived on Zenodo at DOI: 10.5281/zenodo.11105506 (Schaphoff et al., 2024) under the AGPLv3 license. The exact version of the model used to produce the results used in this paper is archived on Zenodo at DOI:

10.5281/zenodo.11473451, along with the MATLAB implementation of the Rothermel model, and input data and scripts used to produce the plots for all of the simulations presented in this paper.

*Author contributions.* LO conceptualized the new validation methods and implemented the prescribed firestarts validation together with WvB and MD. Analysis of literature and its relevance to the SPITFIRE model was conducted by LO with support from MB, WvB, MD, MF, SB, JH, and KT. JH and JRB performed data curation on the input data for the European model runs. Model code for LPJmL-SPITFIRE1.9

was written by WvB, LO, MB, and MD. LO prepared the original draft of this work. All authors participated in discussions and manuscript



revisions. Additionally, MB performed the model tuning for the European model version, led the development of the phenology-based live grass moisture parametrization, and prepared Figure 6.

*Competing interests.* The authors declare no competing interests

*Acknowledgements.* This research was performed within the research training group "Natural Hazards and Risks in a Changing World"
(NatRiskChange) funded by the Deutsche Forschungsgemeinschaft (DFG; GRK 2043/2). This project has received funding from research project FirEUrisk, a European Union Horizon 2020 research and innovation program under Grant Agreement No. 101003890. The authors gratefully acknowledge the European Regional Development Fund (ERDF), the German Federal Ministry of Education and Research and the Land Brandenburg for supporting this project by providing resources on the high performance computer system at the Potsdam Institute for Climate Impact Research. The authors thank Dr. Christoph Müller, Dr. Susanne Rolinski, and Dr. Sibyll Schaphoff for the development
and continued maintenance of LPJmL version 5.

## Appendix A: Detailed errors in the SPITFIRE implementation of the Rothermel Model

### A1 Background on the Rothermel equation

The Rothermel equation, Equation A1, is a semi-empirical rate of spread equation first developed by Rothermel (1972). Fundamentally, it derives the rate of spread of a fire by dividing the rate at which energy is released towards a section of fuel by
the amount of energy that is required for that section of fuel to ignite. In detail:

$$R = \frac{I_R \xi (1 + \phi_w + \phi_s)}{\rho_b \epsilon Q_{ig}}, \tag{A1}$$

where $R$ is the rate of spread, in m/s, $I_R$ is the reaction intensity, i.e. the rate at which energy is released by the fire, in $\mathrm{W\,m^{-2}}$, $\xi$ is the propagating flux ratio, the proportion of that energy that is transmitted to the subsequent part of the fuel bed, and $\phi_w$ and $\phi_s$ are factors that describe the effect of wind and slope on the propagating energy flux. In the denominator, $\rho_b$
is the bulk density of the fuel, in $\mathrm{kg\,m^{-3}}$, $\epsilon$ is the effective heating number, i.e. the proportion of a fuel particle that must be heated to ignition for combustion to occur, and $Q_{ig}$ is the amount of energy required to heat a kilogram of fuel to ignition, in $\mathrm{J\,kg^{-1}}$ (Rothermel, 1972; Andrews, 2018).

The individual parameters of Equation A1 are functions of further fuel bed parameters, including the fuel bed depth, surface-area-to-volume ratio of the fuel particles, fuel moisture content, fuel moisture of extinction and heat content. In the case of a
uniform fuel bed, the surface-area-to-volume ratio and moisture content implemented in the Rothermel equation are simply the moisture content and surface-area-to-volume ratio of the uniform fuel particle size. In the case of a non-uniform fuel bed, Rothermel (1972) introduced a weighting system based on the contribution of individual fuel size classes to the overall surface area of the fuel bed. The commonly used formulation of this weighting system is the updated version described in Albini



(1976). In this system, the fuel bed is divided into living and dead fuel categories as well as different fuel size classes. Each category and size class has a weighting factor, $f_{ij}$, assigned to it, where $i$ connotes the living and dead fuel categories and $j$ connotes the different fuel size classes. These factors are calculated using:

$$f_{ij} = \frac{\sigma_{ij} w_{0,ij}/\rho_{p,ij}}{\sum_j \sigma_{ij} w_{0,ij}/\rho_{p,ij}}, \tag{A2}$$

where $\sigma_{ij}$ is the surface-area-to-volume ratio of a fuel class, in $\mathrm{m}^{-1}$, $w_{0,ij}$ is the oven dry fuel load of the fuel class, in $\mathrm{kg/m}^2$, and $\rho_{p,ij}$ is the particle density of the fuel in the fuel class, in $\mathrm{kg\,m}^{-3}$. The summation in the denominator signifies a summation over all components within the dead or alive fuel category, $i$. The numerator in Equation A2, therefore, is the total surface area of a given fuel class in a unit area of the fuel bed, and the denominator is the total surface area of all classes over that same unit area. This results in a weighting by fuel surface area. As an example, the combined, representative surface-area-to-volume ratio of the dead fuel category ($i = 1$) would be:

$$\sigma_1 = \sum_j f_{1,j} \sigma_{1,j}, \tag{A3}$$

One subtlety, added to this approach by Albini (1976), is the inclusion of the $g_{ij}$ weighting factors for combining fuel loads. These factors are similar to the $f_{ij}$ factors, with the distinction that they gather the individual fuel classes into bins based on their surface-area-to-volume ratio and assign a single weighting factor to all fuel classes in a bin. This circumvents a conceptual issue in the original Rothermel equation where the $f_{ij}$ factors result in differing combined fuel loads depending on how the fuel classes are partitioned (Albini, 1976; Andrews, 2018). The g factors for each size bin are calculated using:

$$g_{ij} = \begin{cases} \sum_{\mathrm{bin}} f_{ij}, & \text{if } \sigma \geq 0.54\,\mathrm{cm}^{-1} \\ 0, & \text{otherwise} \end{cases} \tag{A4}$$

## A2 Implementation of the Rothermel equation in SPITFIRE

Fundamentally, the main error in the implementation of the Rothermel equation in SPITFIRE results from an application of the Rothermel equation for a uniform fuel bed to the non-uniform fuel beds present in the vegetation models with which SPITFIRE operates. Rather than applying the form designed for a non-uniform fuel bed, the variance in fuel bed parameters is accounted for by combining them according to an incorrect weighting scheme. In this scheme, the surface area-based weighting factors were neglected in the implementation of the Rothermel equation and a different weighting approach was taken, whereby most of the components are weighted by their contribution to the total fuel bed mass, e.g.:

$$\sigma = \sum_i \sum_j \sigma_{ij} \frac{w_{0,ij}}{w_0}, \tag{A5}$$





where $w_0$ is the total oven-dry fuel load of the fuel bed (Thonicke et al., 2010). The standard weighting system shown in
Equation A2, due to its weighting by contribution of each fuel class to the total surface area of the fuel bed, is more heavily
weighted toward fine fuels. The SPITFIRE weighting system in Equation A5, however, is more heavily weighted towards
coarse fuels as they are heavier and therefore contribute to a greater proportion of the overall fuel bed's mass.

In addition to the lack of $f_{ij}$ factors in SPITFIRE, the $g_{ij}$ factors are omitted entirely, and the fuel load that is inserted into
the Rothermel equation is a simple sum of the individual fuel loads:

$$w_0 = \sum_i \sum_j w_{0,ij}, \tag{A6}$$

(Thonicke et al., 2010). This contributes an even larger source of error since the replacement of a weighted average by a
simple sum can increase the fuel load by a factor up to the number of individual fuel components there are (e.g. if there are 3
fuel classes the sum may be 3 times larger than the average).

Further, an element of the Rothermel equation that is lacking from SPITFIRE is the separate treatment of dead and live fuels.
For example, the reaction intensity in the non-uniform fuel bed Rothermel equation is calculated using:

$$I_R = \Gamma' \sum_i w_{0,i}(1 - S_{T,ij})h_i\eta_{M,i}\eta_{s,i}, \tag{A7}$$

where $\Gamma'$ is the optimum reaction velocity of the fuel, in $\min^{-1}$, $S_{T,ij}$ is the mineral content of the fuel component, $h_i$ is the
weighted average heat content of the fuel bed component, in $\text{kJ kg}^{-1}$, $\eta_{M,i}$ is a moisture dampening coefficient and $\eta_{s,i}$ is a
mineral damping coefficient (Andrews, 2018; Albini, 1976). The moisture damping coefficient is calculated using:

$$\eta_{M,i} = 1 - 2.59\frac{M_{f,i}}{M_{x,i}} + 5.11\left(\frac{M_{f,i}}{M_{x,i}}\right)^2 - 3.52\left(\frac{M_{f,i}}{M_{x,i}}\right)^3, \tag{A8}$$

where $M_{f,i}$ is the moisture content of a fuel component, in kg water per kg oven dry mass, and $M_{x,i}$ is the moisture
of extinction of that fuel component. As shown in Equations A7 and A8, the moisture contents of the dead and alive fuel
components are treated separately before being combined in the Rothermel equation. In the implementation of the Rothermel
Equation in SPITFIRE, however, the moisture contents of the dead and live fuel components are combined using the following
series of equations:

$$\alpha_2 = -ln(M_{f,2})/\text{NI} \tag{A9}$$

$$M_{f,combined} = \exp(-1 \times (\alpha_1 \frac{w_{0,1}}{w_0} + \alpha_2 \frac{w_{0,2}}{w_0}) \times \text{NI}), \tag{A10}$$

where NI is the Nesterov Index (see Thonicke et al., 2010). Note these equations are present in the SPITFIRE code, docu-
mented in Schaphoff et al. (2018b), and alluded to in Pfeiffer et al. (2013). However, they are not explicitly given in Thonicke
et al. (2010). Equations 8 and 9, together with Equation 6 in Thonicke et al. (2010), can simply be combined into a single
equation:

$$M_{f,combined} = M_{f,1}^{\frac{w_{0,1}}{w_0}} M_{f,2}^{\frac{w_{0,2}}{w_0}}. \tag{A11}$$

Effectively, SPITFIRE uses a combined moisture content that is a weighted geometric mean of the dead and live fuel moisture
contents and places this in the Rothermel equation for uniform fuels, despite operating on non-uniform fuel beds.





## Appendix B: Sources of uncertainty in specific model components

### B1 Ignitions

#### Human ignitions

Human ignitions in SPITFIRE are calculated using a simple function of population density and a parameter that describes the propensity of humans in a given grid cell to cause ignitions. The full equation is (combining Equations 3 and 4 in Thonicke et al. (2010))

$$n_{h,ig} = 30 P_D a(N_D) e^{-0.5 \times \sqrt{P_D}} / 100, \tag{B1}$$

where $P_D$ is the population density, individuals per km$^2$, and $a(N_D)$, ignitions per individual per day, is the propensity of people in a given grid cell to cause ignitions. This function was based on theoretical considerations, including findings by Archibald et al. (2009) for southern Africa, that suggest a relationship between population density and the number of ignitions that increases until reaching a maximum at intermediate population densities, followed by a decrease. The $a(N_D)$ parameter was derived for some regions using local fire databases and used as a tuning parameter for others. This parametrization does not explicitly take into account different fire management practices, and how these practices depend on local conditions. Connecting this parametrization to these measures, as has been explored, e.g., by Perkins et al. (2022), is an avenue of future model development that may improve burnt area distribution in SPITFIRE.

#### Lightning ignitions

The five years of the LIS/OTD monthly data set used in SPITFIRE were averaged in the original parametrization to produce one year of monthly values that was then interpolated to a daily lightning data set. This results in a consistent, low amount of lightning ignitions, with a small fraction of an ignition on most days. Because of this, the lightning ignitions are also de-synchronized from any precipitation that may accompany them. The impact of temporal resolution on lightning ignitions has also been explored by Felsberg et al. (2018), showing a small impact on a global scale but greater effects regionally. Alternative lightning ignition parametrizations for use in SPITFIRE-related models have been suggested by Pfeiffer et al. (2013) and Kelley et al. (2014) that may help to remedy this. Because these models are SPITFIRE-related, testing their implementations in future versions of SPITFIRE may be straightforward. Further model development may also be based on newer lightning datasets, e.g. Kaplan and Lau (2021), that allow for a better understanding of inter-annual variability in lightning ignitions and their impact on burnt area. Given the new datasets available, and the improved model foundations here, revisiting the lightning parametrization in SPITFIRE may be a fruitful area of research, particularly for regions that are remote from human activity and in which lightning is, therefore, the dominant source of ignitions.

### B2 Fire spread

#### Surface-area-to-volume ratio





The SPITFIRE model undertakes some simplifications in its fuel bed parameters that may be built upon in future work. This includes uniform surface-area-to-volume ratios for fuels across all vegetation types. While this is common for the coarser size classes, the surface-area-to-volume ratios of the 1-hour fuel classes generally show some distinction between e.g. leaf-dominated fuel beds and needle-dominated fuel beds. The Scott and Burgan (2005) fuel models, for example, have 1-hour surface-area-to-volume ratios that range from 1500 to 2200 ft$^{-1}$ (49.2 to 72.2 cm$^{-1}$), not including shrub fuel models, whereas

SPITFIRE uses a uniform value of 2021 ft$^{-1}$ (66 cm$^{-1}$). Since the surface-area-to-volume ratio is a key parameter for calculating many of the variables in the Rothermel model, potential model improvements may be made by varying the 1-hour surface-area-to-volume ratio for different vegetation types.

**Dead fuel moisture of extinction**

       The dead fuel moisture of extinction in SPITFIRE is a uniform value of 30 %. This is distinct from, e.g., the Scott and

Burgan (2005) fuel models, where $M_{x,\mathrm{dead}}$ has a range of 15-40 %, with values depending on the type of vegetation and climate. Therefore, variable dead fuel moistures of extinction are another potential improvement for the SPITFIRE model. A challenge in this regard would be determining an appropriate manner for constructing a cross-walk between the Scott and Burgan (2005) fuel models and the LPJmL PFTs as the fuel models depend on both vegetation and climatic parameters.

**Crown and ground fires**

Future model developments may expand the fire spread component of the SPITFIRE model to include other forms of fire behaviour. Currently, the model only models surface fire spread, but crown fires and ground fires are also major contributors to fire dynamics. Crown fires have been incorporated into DGVMs previously by, e.g., Ward et al. (2018), and are important due to the increased vegetation mortality they may cause, and high spread rates present in them. Ground fires, among other factors, play a substantial role in arctic fire dynamics and may therefore be relevant to future modelling endeavours that focus on fires

at high latitudes (McCarty et al., 2021).

**Convective and terrain-driven fires**

       Another dynamic that may be relevant for future inclusion in the SPITFIRE model is the effect of atmospheric stability on fire growth and the potential for larger and more severe fires during extreme pyroconvection events (e.g., Senande-Rivera et al., 2022). In addition, terrain can act to promote fire spread through increased rates of spread when fires are spreading up slopes

(although, at a grid cell scale this effect may be counteracted somewhat through the barriers to fire spread that complex terrain creates).

**Fire suppression**

       Finally, a dynamic that can impact the incidence and spread of fires, and is currently not included in the model, but may improve modelling accuracy in future, is the impact of fire suppression on fire size and incidence.



## Appendix C: The impact of model resolution

### C1 Spatial resolution

Wildland fires are an inherently multi-scale phenomenon, with conductive heat transfer that can occur on the scale of microns up to fire-atmosphere interactions that can occur on scales of tens of kilometers (e.g., Collin et al., 2011; Potter, 2012). Because of this it is often necessary to include empirically-derived equations in process-based models to account for sub-grid-cell processes. In the case of SPITFIRE, a substantial factor that is not currently accounted for is the arrangement of fuel beds at a sub-grid level. Fuel bed heterogeneity may arise from terrain, the spatial arrangement of vegetation, and human influence, and it can have strong effects on determining the final shape and size of fires (e.g., Sharples, 2009; Brown, 1981; Narayanaraj and Wimberly, 2011). We therefore highlight this as an area of potential future model development.

The spatial arrangement of fuel beds can also be influenced by previous fires, and this leads to an issue in LPJmL-SPITFIRE specifically due to the manner in which fuel consumption is calculated. In the case of fuel consumption, the current approach in the model is to apply fuel consumption to the entire grid-cell-averaged fuel bed, scaled by the burnt area. Because of this if, e.g., 5 % of the grid cell is burnt with sufficient intensity to consume the entirety of the fuel, the model treats this as 5 % of the total grid cell fuel bed being "skimmed off." This can substantially weaken fuel feedbacks in grid cells where a low portion of the grid cell is burnt, due to the small reduction in fuel load. In LPJ-GUESS-SPITFIRE this issue is mitigated by including burnt patches as separate stands, allowing for strong, local effects of fires. This approach may be an avenue for improvement of other SPITFIRE implementations, and it may be preferable for users of SPITFIRE to work with an implementation in LPJ-GUESS if these feedbacks are important to their use case.

### C2 Temporal resolution

SPITFIRE operates on a daily time step and many processes in the DGVMs with which it is coupled generally operate on an annual time step. Processes on the daily time step generally use daily-averaged values to calculate parameters such as the dead fuel moisture content. Because fire spread largely occurs during the day (e.g., Balch et al., 2022), there may be some bias in applying daily averaged values as inputs and model accuracy may be improved in future by considering inputs on the basis of the times of day during which fire spread occurs. For example, the 1 h dead fuel moisture is highly subject to sub-daily variations and may be better captured if calculated using daytime values.

Regarding the time step of DGVMs, in LPJmL, for example, while mortality due to fire is calculated on a daily time step, the background mortality and establishment of vegetation occur on an annual time step. This is particularly relevant for fires in the southern hemisphere or areas in the tropics where fire spread occurs at the end of the year (see, e.g., Giglio et al., 2013), because it can result in fires on December 31st that experience substantially less standing vegetation and lower litter loads than fires on January 1st. In regions where depleted fuel beds are regenerated by a monsoon season, and that therefore show a bimodal fire season (Archibald et al., 2009), the annual establishment time step may result in substantially under-loaded fuel beds in the second half of the year relative to their real-world counterparts. Therefore, this annual time step may impact the seasonality of fires in many parts of the world and should be considered if fire seasonality is a desired model output. Future



work in this direction may improve this issue by spreading mortality and establishment around the year, rather than confining them to the year's end.

## C3  Division of global vegetation into PFTs

The final manner in which model resolution impacts results is the division of global vegetation into individual PFTs. This is necessary as it is infeasible to model every species and sub-species difference of vegetation globally. However, this can pose challenges, particularly since the PFTs in DGVMs are often not chosen to reflect the differences in their response to fire (e.g. Fischer et al., 2018). For example, the Boreal Needleleaved Evergreen PFT represents a wide variety of tree species including, e.g., spruces and pines. Because spruces generally have a fundamentally different structure, with branches that extend much closer to the ground, they are more likely to experience crown scorch (e.g., Lacand et al., 2023). Further research into fire-specific PFTs may include studies of specific vegetation adaptations to fire as discussed, e.g., by Harrison et al. (2021), or the inclusion of shrub PFTs in DGVMs that do not include them, since shrubs can be an important factor in fire and post-fire dynamics (e.g. Baudena et al., 2020). To allow for the inclusion of shrubs in future we have included a, currently empty, live woody fuel category in our implementation of the Rothermel model so that shrub parameters can simply be input into the fire spread calculations.

Another example key to LPJmL-SPITFIRE is the uniform tree height and diameter for a given PFT in a given grid cell that the LPJmL DGVM operates with (Schaphoff et al., 2018a). When combined with the tree-height and -diameter based mortality functions of SPITFIRE, this poses a challenge in accurately representing the mortality of vegetation that can be quite hetero-geneously sized in actual fact, but is uniform in the model. Future developments in DGVMs that allow for heterogeneous tree sizes, as has been incorporated into LPJ-GUESS, may reduce the uncertainty in this regard, and the importance of vegetation heterogeneity should be considered when applying SPITFIRE to a desired use case.



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
