# Peer review of "Sources of Uncertainty in the Global Fire Model SPITFIRE: Development of LPJmL-SPITFIRE1.9 and Directions for Future Improvements"

_EGUsphere, 2024_

## Author Comment (AC1)

We thank the reviewers for their thoughtful and constructive comments. We are pleased by both reviewers' positive assessments of the scientific significance, quality, and reproducibility of the manuscript. We have addressed the concerns of Reviewer #1 regarding the organization of the manuscript, and all other comments, below. We believe that the comments classified as needing major revisions, which generally refer to these organizational concerns, can be resolved by revisions containing additional points of clarification and adjusting the location of some text and figures, as described in detail below. The reviewers' comments are in italics, our responses are in regular font, and new text to be included in the manuscript is shown in blue.

**Comments from Reviewer #1**

*The authors of Sources of Uncertainty in the Global Fire Model SPITFIRE: Development of LPJmL-SPITFIRE1.9 and Directions for Future Improvements present an overview of the SPITFIRE model. They review the underlying theoretical basis of the model and attempt to better understand sources of error and model uncertainty. They make several revisions to the model to improve model performance and then evaluate these changes across the European modeling domain. The work appears rigorous from a technical and methodological standpoint. It will be of interest to both active users of the SPITFIRE model, as well as users of other fire models interested in incorporating similar improvements of carrying out model intercomparison.*

We thank the reviewer for these positive comments about the technical content of the manuscript and its relevance to the community.

*Major points:*

- *The methods section is rather short as is the description of the model, the paper moves quickly into the result and, there are very few equations presented and a lot of references to model description appendices. This hurts the flow and clarity of the sections where the underlying theoretical basis is being reviewed especially for readers working with other fire models. Some examples include near line 105 where the methods are presented parameters, and values names, are difficult to follow. Near line 175 the prescribed fire starts could be more completely explained. I believe there are numerous other examples of this throughout the text. Getting information out of the appendices and into the main text would improve overall clarity and readability.*

We thank the reviewer for these comments and have implemented them as described below. Regarding the organization of the manuscript in general, our aim was to make the manuscript accessible for two groups of readers who may not read the paper sequentially. The first group is those readers who are interested in specific components of the model. To improve the ease of finding information for these readers, we chose to keep many of the equations for each model component in the results section along with our findings. We feel that this prevents readers from having to flip back and forth in the paper when reading about a specific section. This may also help readers who are reading the paper sequentially, because they do not have to remember highly specific information from much earlier sections. Particularly since the different results sections cover material from somewhat disparate areas of wildfire theory.

The second group of readers, which may overlap somewhat with the first, is readers who are interested in the subject matter described, but for whom a very high level of granularity in some parts may be distracting. For example, this may include readers who

are looking to understand the issues in the fire spread component of the model, but are not looking for a breakdown of the large number of equations involved. We feel that placing this information in an appendix helps retain the necessary information while keeping the manuscript accessible to a broader audience.

However, we acknowledge that there are adjustments that can be made to the manuscript to improve its flow and clarity for those reading it from beginning to end, and we thank the reviewer for the recommendations that will improve the manuscript in this regard. To improve the clarity of the manuscript, we propose adding a paragraph to the introduction that makes the organization of the manuscript explicit, together with moving Figure 8 as suggested in one of the reviewer's subsequent comments. Regarding moving information out of the appendices, we have done so in response to some of the subsequent comments. The new paragraph at the end of the introduction (from line 69) will read:

"The present work is organized into a brief methods section, Section 2, giving a general description of the SPITFIRE model as well as the forcing data used, two new validation methods we have developed, and the model runs at the European scale that we have performed to test these developments. The results and discussion section is organized into a discussion of the two major issues that were identified in SPITFIRE, followed by further discussions of individual model components, results of model tests in the European domain, and a section describing the current status of the model. The sections describing specific model developments and potential future developments are summarized in Figure 1 for readers who are interested in individual developments or model components."

Figure 1 in this paragraph refers to the previous Figure 8. All figures have been renumbered accordingly.

In addition to this change, we propose clarifying the description of the FDI near line 105, replacing lines 101 – 106 with:

"Subsequently, in component 2, the number of ignitions is scaled using a Fire Danger Index (FDI) with a value between 0 and 1. This index captures the proportion of ignitions that successfully become spreading fires. There are three versions of this FDI that have been implemented in different versions of SPITFIRE, shown in grey in Figure 2. All versions aim to account for the role of moisture and fuel bed composition in determining the likelihood of a spreading fire following a potential ignition event.

The original version of the FDI, described in Thonicke et al. (2010), is a function of a previously existing fire danger index, the Nesterov index. The Nesterov index is calculated cumulatively over days in which there are less than 3 mm of precipitation, and is a function of daily maximum temperature and dew point temperature. A weighted-average relative moisture content of the fuels being burnt is then calculated based on an exponential function of the Nesterov index and the geometry of the particles making up the fuel bed. The ratio of this moisture content to the moisture of extinction is used as the FDI."

We propose augmenting the explanation of the prescribed firestarts results by replacing the sentence from lines 174-176 with the following text:

"Andela et al. (2019) identify this region as having a high ignition density and small fire sizes (see their Figure 8). That a high ignition density is required to produce appreciable burnt area in LPJmL4-SPITFIRE suggests that individual fires in the model are quite small and, therefore, a large number of them is required. Therefore, the validation using prescribed firestarts offers additional insight into the model performance that was not available from the original burnt area maps."

- *There are also some sections (i.e. near line 545 and table 1) where there is information presented quite late in the paper in the results section that seems more methodological. Some re-organization here and elsewhere for clarity. Having this information presented earlier would improve the readability of the manuscript.*

For the section describing the European model version, we agree that some information can be moved to the methods section, as it is more general in nature than the sections before, and thank the reviewer for this improvement. We propose adding a new subsection, 2.4, to the methods section to include this information. This section will include the information from the first paragraph of Section 3.3, starting at line 545. The new part of the methods section will read:

"To examine the impact of our model updates, we create a preliminary model version specifically for the European domain. We choose this area as a test case because we aim to restrict the amount of variability the model has to account for on a global scale and due to the involvement of the SPITFIRE model in the FirEUrisk project (https://fireurisk.eu). The new model version uses data available through the FireEUrisk project at a 0.07° grid cell resolution, also allowing for less sub-grid variability than the usual 0.5° scale. A full new version of the SPITFIRE model is reserved for further work pending additional testing and operation on a global scale. We designate this updated model version as LPJmL-SPITFIRE1.9, reserving the label LPJmL-SPITFIRE2.0 for a version that has been tested at the global scale.

We compare the results of the new model version with the standard SPITFIRE model. For this comparison we implement both model versions in LPJmL5.7, with its included litter moisture (developed by Lutz et al., 2019, and described in detail in Section 3.2.5). To allow for a direct comparison of the fire spread and mortality processes, we apply the new tuning parameters and the multi-day fire spread that we develop in this work to the old model version as well. In all other respects, the new model version contains the new improvements and the old model version does not. We then analyze the differences in burnt area and rate of spread between the two model versions, also using the PFT-split burnt area validation method. Note that due to the lack of a prescribed firestarts input at the 0.07° scale, we do not perform the prescribed ignitions validation for this smaller scale version, and reserve such tests for future larger scale versions in which the preliminary fire duration function we develop in this work can be further updated as well."

- *Figure 8: This figure may not be referenced in the text. It also appears very late in the paper. But there are still some results sections that follow it. Organizing these sections to be a bit later in the text or making use of this figure earlier on could improve the manuscript.*

We thank the reviewer for this suggestion, and agree that Figure 8 could be better placed earlier in the manuscript. We propose making it Figure 1 and moving it to the end of the Introduction section as described in the response to the first comment, in addition to the existing brief reference to it at the start of Section 3.4.

- *Line 403: Do the authors mean "inter-specific" here? PFTs tend to be coarse and wouldn't represent differences between species. "Intra-specific" differences and "adaptation" would be on an even lower level and a lot of what's in Appendix C3 seems to focus on "inter-specific" differences. Revising this for clarity and moving some text and references from C3 could address this.*

We do mean intra-specific as stated in the text in this case, but will clarify this in the updated manuscript. The two papers we cite, Bristiel et al. (2018) and Keep et al. (2021), both perform experiments on individual species, Dactylis glomerata and Lolium perenne respectively. The division of vegetation into PFTs, therefore, can gloss over both intra- and inter-specific differences. We propose to clarify this in this section by adding the following sentence after the one ending on line 403, including the reference to Fischer et al. (2018) moved from Appendix C3:

"In addition to these intra-specific adaptations, the use of broad PFTs can also result in the loss of inter-specific differences (as discussed by, e.g., Fischer et al., 2018)."

**Minor points:**

- *Abstract: reword "strong upward biases"*

We propose rewording this to "large positive biases"

- *Abstract: reword "moisture parameterization that results in substantial too low live grass moisture contents"*

We propose rewording this to: "a live grass moisture parametrization that results in unrealistically dry grasses"

- *Abstract: suggest revising the sentence stating "that bias SPITFIRE towards higher tree mortality" to better explain the mechanism. Is this tree mortality in grasslands?*

We propose dividing this sentence into two parts, reading:

"The combination of these issues leads to excessively large and intense fires, particularly on the dry modelled grasslands. Because of the tall flames present in these intense fires, which can cause substantial damage to tree crowns, these issues bias SPITFIRE toward a high tree mortality"

- *Line 83: when "assigning a uniform site" clarify if this means vegetation height is static or dynamically determined by the vegetation model.*

We propose clarifying this by changing the wording from "is assigned a uniform size" to "has a dynamically calculated uniform size"

- *Line 613: reword "upward biases"*

We propose rewording this to: "large positive biases"

- *Line 657: add information and reference for regions where shrubs are an important part of the fire regime for clarity.*

We propose adding more detail by moving the reference to Baudena et al. (2020) from Appendix C3 into this section and changing the sentence in Line 657 to:

"A common issue in several DGVMs in which SPITFIRE is integrated is the lack of a shrub PFT for regions in which shrubs are important parts of the fire regime, e.g. chaparral-covered regions of California and parts of the Mediterranean where shrubs pay a key role in post-fire dynamics (Weise et al., 2016; Baudena et al., 2020)."

The new reference to Weise et al., 2016 is to the following paper:

Weise, D. R., Koo, E., Zhou, X., Mahalingam, S., Morandini, F., & Balbi, J.-H. (2016). Fire spread in chaparral – a comparison of laboratory data and model predictions in burning live fuels. International Journal of Wildland Fire. 25(9): 980-994, 25(9), 980–994. https://doi.org/10.1071/WF15177

**Comments from Reviewer #2**

*In this study, Oberhagemann et al. conduct a thorough review of the SPITFIRE fire module, identifying sources of uncertainty and implementation errors, including two major errors in the model. The study also uses two new methods for validating LPJmL-SPITFIRE that allow for improved validation of different components of the fire model that extend beyond the usual comparison to the observed burnt area. The authors implement numerous improvements to the model and evaluate the improved version across a European domain, discussing uncertainties that exist and areas for future model developments. The work appears to use valid approaches and methods and presents important corrections for errors that have consequences for SPITFIRE-coupled models and potentially other fire models that were originally based on the Rothermal fire spread equations. It is therefore of interest to the fire modelling community and more generally the vegetation and earth system modelling community that include SPITFIRE or closely related fire models within.*

We thank the reviewer for these positive comments, and the constructive comments below.

***Specific comments:***

- *The study starts with LPJmL version 4.0, but the improvements and model developments shown later in the study are using LPJmL version 5.7. The authors argue that version 4.0 was used to start as it is the most recently published global version of SPITFIRE, while the more recent version was used to include the most recent updates to LPJmL. Whilst it is understandable to use v4.0 to illustrate the errors and uncertainty in a published version, it would be nice to see a comparison between the model versions themselves if different versions are to be used. For example, how do figures 2 and 3 look using LPJmL v5.7? This would help make the results more comparable throughout the manuscript.*

We will add versions of Figure 2 and 3 created using a global run with LPJmL version 5.7, and with the same version of SPITFIRE as used in LPJmL version 4. One complicating factor is that there is no globally calibrated version of LPJmL5.7-SPITFIRE, and creating one is out of the scope of this work. Because of this, we can only produce an uncalibrated version that results in a very high burnt area. However, these figures support the same conclusions as Figures 2 and 3 regarding the division of burnt area into tree- and grass-dominated cells, and the highly reduced burnt area when using prescribed ignitions. Because of this, we propose adding these figures to the supplement.

- *Throughout the manuscript, where different models and approaches are mentioned, more details on differences would help inform the reader. For example, in line 31: Various DGVMs are listed, including several LPJ-based models. A brief statement on how the models differ from one another or their strengths and weaknesses would be more informative than a list. At least for the LPJ models, since it's often confused what is included in each. Related to this, is the multi-day burning the same that is implemented in the LPJLM-fire DGVM?In line 81, "This description does not apply to other vegetation models in which SPITFIRE is implemented." More detail on why this is the case is needed to describe in which cases the description applies—for instance, is it that other DGVMs do not use the area-averaged approach but patch/cohorts, etc.?*

We propose adding the following clarifying statement to the end of line 31:

"Some key differences between these applications and the original version include updated lightning ignitions (LPX, LPJ-LMfire, and ORCHIDEE), parametrizations for differences in human fire use (LPJ-LMfire), a population density effect on fire duration (JS-BACH), stochastic burning of vegetation patches (LPJ-GUESS), and empirically derived regional scaling of burned area (ORCHIDEE)."

We propose adding the following clarification after the statement that the "description does not apply to other vegetation models…":

"For example, LPJ-GUESS uses a cohort approach with different age classes per PFT, and accounts for stochasticity by simulating multiple patches (e.g. Lehsten et al., 2009; Lehsten et al., 2015). The number and types of PFTs varies between DGVMs as well, for example the JSBACH version used in phase 1 of FireMIP has 12 PFTs, including shrubs (Rabin et al., 2017; Lasslop et al., 2014)."

Regarding the multi-day fire spread algorithm, our implementation differs from the one in LPJLM-fire in a few key ways. First, the implementation in LPJLM-fire retains the original daily fire duration function, with a maximum of 241 minutes and a step-like shape that caused half of all fires to spread for less than 30 minutes per day, an extremely low value (as discussed in the fire duration section of our manuscript). In our implementation, we have changed the daily fire duration function to address this. Second, the criterion for multi-day fires to extinguish in LPJ-LMfire is based solely on accumulated precipitation. Our FDI-based criterion allows for fires to extinguish due to precipitation, but also through other ways in which the fuel moisture may increase, for example due to a drop in temperature that reduces the evaporative demand. Finally, the implementation in LPJ-LMfire includes several additional means of reducing the number of fires as a function of, e.g., the number of actively burning fires. While we agree that this is a good inclusion in principle, we do not feel that there is sufficient analysis

available at the current time to parametrize this for a global model operating on coarse grid cell resolutions, and we have discussed the need for this analysis in Section 3.4. We propose adding the following clarifying statement to the end of line 486:

"The updated multi-day fire spread algorithm also differs from the one implemented in LPJ-LMfire by Pfeiffer et al. (2013) in the duration of fires per day, since Pfeiffer et al. (2013) use the original daily fire duration function, and in the extinction criteria, since we choose a more flexible criterion based on the FDI rather than one that is based only on changes in fuel moisture due to precipitation."

- *For Figure 2/3. It could be useful to have some spatial plots of simulated vegetation, as well as defined grass and tree grid cells, compared to observational to show the vegetative differences and eliminate them as a cause for differences in burnt area. Similarly, in the caption of Figure 2, examination of individual tree PFTs is mentioned but not shown. Whilst these plots may crowd the manuscript, they could be added into the SI.*

We will add these to the supplement as suggested. While there are differences between the simulated vegetation and the validation data, the grid cells that meet our criteria for inclusion, and in which there is substantial fire, fall in the same broad geographic regions. This is especially the case for the grass-dominated grid cells. There is a difference in the extent of tree-dominated grid cells in fire-prone regions that can be attributed to excessive tree mortality due to fire, as shown in the European model runs and the existing Figure S5.

We will also add the plots for individual tree PFTs. In re-examining these, it became clear that the caveat given in the caption of Figure 2, i.e. that the burnt area for tree PFTs only shows good agreement with observations when they are combined, can be explained by a grid misalignment. We have repaired this, and the overall burned area in cells dominated by tree PFTs is now substantially underpredicted, in line with the findings for individual tree PFTs, and we apologize for this oversight. The cause for this under burning in tree-dominated areas is most likely the excessive tree mortality mentioned in the previous paragraph.

We will therefore update panels d) and e) of Figure 2 and their associated descriptions as well (however, the grass results in panel e) remain very similar). This misalignment did not affect any of our other results, including the ones for the European domain, because a different approach to aligning the grids was used.

- *From line 175, my understanding is that LPJmL4-SPITFIRE is able to simulate smaller fires in large numbers well and the fewer but larger fires less well. This is also alluded to later in the paper (e.g., L583). Yet Line 180 seems to contradict this, stating that the incorrect implementation results in unrealistically large and severe fires. Is it that one is true on a global scale, while the other is true for just grasslands? Clarification is needed here.*

The description in line 180 was intended to refer to the fact that, given the same inputs, the implementation of the Rothermel model in SPITFIRE results in fires that are larger than if they were calculated using the correct version of the model. That the actual modelled fires are small is most likely due to the very low fire duration. We thank the reviewer for pointing out this lack of clarity and propose the following clarification to the sentence ending in Line 180:

"Two substantial errors in SPITFIRE are an incorrect weighting of parameters in the Rothermel (1972)-based rate of spread calculation that results in unrealistically severe fires that spread too rapidly, and unrealistically low modelled live grass moistures."

- *In Figure 4 and related discussions, ROS differences under low wind speeds have a large impact on fire size, but even larger differences in ROS result in minimal differences in fire size. This is due to the size of rate of spread difference compared to rate of spread values; however,  further clarity is needed for readers. Either giving an example in text of how the larger differences under high wind speeds result in little impact on fire size or ideally replacing Figure 4. Panels a) and b) with % differences instead.*

We thank the reviewer for helping us clarify this in the text and will replace panels a) and b) with % differences as suggested

- *Why were the parameters and factors in section 3.3 chosen? For example, the minimum fire duration of 2 hours and a maximum of 7*

The parameters in this section were chosen to calibrate the burnt area output from the model. As stated in lines 562-564, because there are no clear, empirically derived values for these coefficients, and our main aim is to compare model versions, we take a fairly liberal approach to the model tuning here. Since both model versions use the same tuning, the comparisons should not be affected by this. There is some support for the maximum of 7 hours in the work by Parisien et al. (2010), in which a 7- or 8- hour daily burning period is used in their simulations. The minimum of 2 hours was chosen with the reasoning that if a fire is able to enter steady state spread, as accounted for by the Fire Danger Index part of the model, it is unlikely to be extinguished immediately. Because of this uncertainty in the values, our aim in this work was to produce a functioning version of the model, while being transparent about the uncertainty that remains. For this reason, we also described the model version as a basis for future developments rather than as being complete.

Parisien, M.-A., Miller, C., Ager, A. A., & Finney, M. A. (2010). Use of artificial landscapes to isolate controls on burn probability. Landscape Ecology, 25(1), 79–93. https://doi.org/10.1007/s10980-009-9398-9

***Technical corrections***

- *Line 73: add the most relevant references on SPITFIRE there.*

Done

- *Figure 3: swap around descriptions of b) and a)*

Done

- *Figure 4: Clarify in the caption that it is SPITFIRE-Rothermal and mention of what the TL3/TU2 fuel classes are (or alter the panel titles to include).*

We will update the first two lines of the caption to read:

"Comparison between outputs of the SPITFIRE implementation of the Rothermel equation and the correct implementation created by Rothermel (1972) and Albini (1976). Comparisons are done for the TU2 and TL3 fuel models, moderate load, humid climate timber-shrub and moderate load conifer litter respectively, of Scott and Burgan (2005)."

- *Line 438: "to rectify this," what is meant by 'this'? Reword for clarification.*

We will reword this to:

"To improve upon these issues, we replace this untested parametrization with a new dead fuel moisture parametrization that makes use of a dynamic, LPJmL-based litter moisture calculation developed by Lutz et al. (2019) (described further in Section 3.2.5). This allows us to replace the Nesterov-based FDI with the VPD-based FDI that was more recently developed for SPTFIRE by Drüke et al. (2019)."

---

## Author Response (AR2)

We thank the reviewers and the editor for their helpful comments throughout the review process. We have made the final technical corrections, updating lines 626 and 628 as requested, and have replaced the first sentence of the caption for Figure S9 with "Comparison of burnt area per grid cell from fires in the Global Fire Atlas to the burnt area per grid cell when starting the same fires in the SPITFIRE version from LPJmL4, but implemented in LPJmL5.7. This supplements Figure 4 in the main text showing LPJmL4 with its own SPITFIRE version."